# PHYSICALLY PLAUSIBLE MULTI-SYSTEM TRAJECTORY GENERATION AND SYMMETRY DISCOVERY

## ABSTRACT

From metronomes to celestial bodies, mechanics underpins how the world evolves in time and space. With consideration of this, a number of recent neural network models leverage inductive biases from classical mechanics to encourage model interpretability and ensure forecasted states are physical. However, in general, these models are designed to capture the dynamics of a single system with fixed physical parameters, from state-space measurements of a known configuration space. In this paper we introduce Symplectic Phase Space GAN (SPS-GAN) which can capture the dynamics of multiple systems, and generalize to unseen physical parameters from. Moreover, SPS-GAN does not require prior knowledge of the system configuration space. In fact, SPS-GAN can discover the configuration space structure of the system from arbitrary measurement types (e.g., state-space measurements, video frames). To achieve physically plausible generation, we introduce a novel architecture which embeds a Hamiltonian neural network recurrent module in a conditional GAN backbone. To discover the structure of the configuration space, we optimize the conditional time-series GAN objective with an additional physically motivated term to encourages a sparse representation of the configuration space. We demonstrate the utility of SPS-GAN for trajectory prediction, video generation and symmetry discovery. Our approach captures multiple systems and achieves performance on par with supervised models designed for single systems.

## 1 INTRODUCTION

Understanding and predicting the motion of dynamical systems can offer insights to a wide range of engineering disciplines such as dynamics and control (Van Pelt & Bernstein, 2001; Khalil & Grizzle, 2002), autonomous driving (Lefèvre et al., 2014), and quantum mechanics (Huang et al., 2023). Traditional system identification techniques require the analytical form of the underlying dynamics (Magal & Webb, 2018; Galioto & Gorodetsky, 2020a;b), general functional form of the class of systems being studied (Epperlein et al., 2015; Paredes & Bernstein, 2021; Paredes et al., 2024; Richards et al., 2024), or curated libraries of nonlinear candidate functions (Brunton et al., 2016) in order to accurately model their behaviour.

Early recurrent models (Rumelhart et al., 1985; Jordan, 1997) offered a general framework to model complex systems by trading structure and interpretability for expressivity. Recent works tackle this challenge using physical laws as inductive biases to constrain networks for better interpretability and accuracy (Greydanus et al., 2019; Toth et al., 2019; Chen et al., 2019; Zhong et al., 2019; Cranmer et al., 2020). One approach is to learn the system Lagrangian or Hamiltonians parametrised by a neural network. This approach ensures that the predicted motion satisfies conservation of total energy. Researchers have also been able to integrate these physical inductive biases to predict (Allen-Blanchette et al., 2020) or generate (Toth et al., 2019) physically plausible videos of dynamical system for tasks such as control (Zhong & Leonard, 2020) or anomaly detection (Mason et al., 2023). While these methods significantly reduced the amount structural knowledge we need when modelling mechanical systems, they still assume access to domain specific information such as the dimension of generalised coordinates and number of constraints.

Learning the symmetries or constrains of dynamical system is not a trivial task, as there is no inherent link between the number of symmetries and the observed behaviour. Some existing works attempt to identify system symmetries by sweeping across constants of motion (Kasim & Lim, 2022) or

first derivatives (Matsubara & Yaguchi, 2022). Others focus on identifying conservation in terms of Lie group structure (Lishkova et al., 2023) or by statistical correlation on manifolds (Lu et al., 2023). While these approaches produce interpretable results for toy examples, they don't provide a generalizable framework for identifying physical symmetries.

In this work, we propose Symplectic Phase Space GAN (SPS-GAN) for generating physically plausible trajectories for multiple systems and discovering system symmetries without the need for a priori information. SPS-GAN is designed on a condition GAN (Goodfellow et al., 2014; Mirza & Osindero, 2014) backbone with the latent space dynamics governed by a Hamiltonian Neural Network Greydanus et al. (2019) which allows us to generate physically consistent latent trajectories. We incorporate a cyclic coordinate loss function in the GAN objective to encourage identification of system symmetries. SPS-GAN is able to minimise the dimension learned phase space and identify constraints. We demonstrate that SPS-GAN is able to achieve predictive performance on par with supervised models; identify the number of degrees of freedom for a wide range of dynamical systems, both ordered and chaotic; generate physically plausible videos with more consistency than baseline physics informed methods; and generate video of systems with unseen parameters.

## 2 RELATED WORKS

**Conservation Constraints in Neural Networks.** Hamiltonian formalisms have been introduced as inductive biases in networks to improve the interpretability and accuracy when modelling dynamical systems. Most notably, Hamiltonian Neural Network (HNN) (Greydanus et al., 2019) learns the Hamiltonian of a system using a black box model and predicts the dynamics using Hamilton's equation. Symplectic Recurrent Neural Network (Chen et al., 2019) leverages symplectic integration to resolve energy drift when integrating the Hamiltonian with non-conservative schemes. Further developments leverages insights such as energy shaping to learn physical properties of systems (Zhong et al., 2019); the port-Hamilton formulation to model systems with energy dissipation (Zhong et al., 2020); generating functions to learn symplectic evolution maps in discrete time (Chen & Tao, 2021); soft constraints to define a symplectic map based on the initial energy of a system (Mattheakis et al., 2022); and Riemann geometry to increase the structure of the learned Hamiltonian (Aboussalah & Eddib, 2025). Hamiltonian inductive bias in neural networks has also demonstrated an ability to model systems in high-dimensional representations, such as images. For instance, Hamiltonian rollouts can be utilised to predict video outputs of freely rotating rigid bodies in aerospace applications (Mason et al., 2022; 2023) or unconditionally generate physically plausible videos (Toth et al., 2019; Allen-Blanchette, 2024). These proposed methods have shown to be extremely capable in achieving high predictive performance whilst conserving system energy.

Pivoting to Lagrangian mechanics, Deep Lagrangian Network (Lutter et al., 2019) offers a framework to learn dynamics of systems from arbitrary coordinates, instead of prescribed position and momentum vectors required to compute the Hamiltonian. Lagrangian Neural Network (Cranmer et al., 2020) extends this concept by removing the restriction on the functional form of the Lagrangian, thereby expanding breadth of systems it is able to model. Lagrangian-based methods can similarly extend to higher-order data, using images to learn energy-based control (Zhong & Leonard, 2020) and predict long-term behaviour (Allen-Blanchette et al., 2020). While the Lagrangian is parametrised by easier to access quantities (e.g. velocity instead of momentum), it creates additional computational challenges by requiring Jacobian and Hessian operations to define the Euler-Lagrange equation. Symplectic Discrete Lagrangian Neural Networks (Lishkova et al., 2023) circumvents this by calculating a discrete Lagrangian with analytical gradients from discrete observations.

**Generating Physically Plausible Data.** Several prior works have attempted to construct models to synthesise physically consistent video, leveraging generative models such as VAEs (Kingma & Welling, 2013) and GANs (Goodfellow et al., 2014). Hamiltonian Generative Network (HGN) proposed in Toth et al. (2019) represented a first approach that was able to learn Hamiltonian dynamics from high-dimensional observations such as images. HGN encodes a sequence of images to latent representations and unrolls the dynamics on the latent space using a symplectic integrator subject to Hamilton's equations. However, HGN makes the non-trivial assumption that the dimension of generalised coordinates of the system in question is known a priori. The authors of Gordon & Parde (2021) explore replacing LSTMs (Hochreiter & Schmidhuber, 1997) with NeuralODEs (Chen et al., 2018) in popular GAN based image sequence generation architectures. While it can effectively enhance the expressivity of GAN models, the latent dynamics remain unstructured. Hamiltonian

Latent Operators (HALO) proposed in Khan & Storkey (2022) encodes sequences of images into a collection of Hamiltonian neural operators. It aims to differential between content (which remains constant throughout the sequence) and motion (which evolves in time) which allows the separation of dynamics from content when generating new trajectories. Most similar to ours is HGAN (Allen-Blanchette, 2024), which also uses a HNN recurrent module and sparsity inducing loss but is only able to generate videos of a single systems with fixed parameters at a time.

**Learning Symmetry in Physical Systems.** Symmetries represents a systemic way for practitioners to enforce conservation laws (Matsubara & Yaguchi, 2022) or improve deep learning model generalisation (Yang et al., 2024b; Zhong & Allen-Blanchette, 2025). However, identifying symmetries within dynamics systems represents a non-trivial challenge as this information is often latent to the observed dynamics. SymDLNN (Lishkova et al., 2023) approaches this problem by leveraging discrete Lagrangian formulations, which enables identifying sub-group which acts by symmetries for a given Lie group action on the configuration space. COMET (Kasim & Lim, 2022) leverages the concept of constants of motion to identify conserved coordinates. However, COMET relies on sweeping across the amount of constants of motion to understand the symmetry within the dynamical system. Alternatively, FINDE (Matsubara & Yaguchi, 2022) builds upon the NeuralODE (Chen et al., 2018) backbone to identify and preserve first integrals, but also relies on hyperparameters to bound its latent space. More recently, Latent MOS (Li et al., 2025) achieves more granularity in the discovered symmetries by embedding specific symmetry groups (such as rotation and translation) as a Mixture-of-Experts on the latent space. The authors of Lu et al. (2023) move away from parametric models and use manifold learning techniques to identify conservation laws by focusing on the geometry of the phase space. WSINDy (Messenger et al., 2024) applies coarse-graining capability to the setting of Hamiltonian dynamics to reveal approximate symmetries associated with timescale separation, but domain knowledge is required to select appropriate basis functions.

## 3 BACKGROUND

In this section we discuss Hamilton's equations and describe how they can be used to identify cyclic coordinates. We also include a brief discussion of Hamiltonian Neural Networks (Greydanus et al., 2019; Chen et al., 2019) as it determines the dynamics of the latent space of SPS-GAN.

### 3.1 HAMILTONIAN MECHANICS

Hamiltonian mechanics reformulates Newton's second law in terms of the energy, rather than the forces. Hamilton's principle states that dynamical systems move along a path that minimises the difference between kinetic and potential energy. This constraint provides a general way of dealing with complex mechanics (Marion, 2013). The Hamiltonian $\mathcal{H}(q, p)$ of a system maps the generalised position $q$ and momentum $p$ to the total energy. The behaviour of such as system is then governed by Hamilton's equation

$$\dot{q} = \frac{\partial \mathcal{H}}{\partial p}, \qquad \dot{p} = -\frac{\partial \mathcal{H}}{\partial q}. \tag{1}$$

In this formulation, it is clear that if a coordinate $q$ does not contribute to the energy we have

$$\dot{p}_k = -\frac{\partial \mathcal{H}}{\partial q_k} = 0. \tag{2}$$

Coordinates $q$ which satisfy this equation are called cyclic or ignorable coordinates. This constraint is used in Section 4.3 to define the cyclic coordinate loss for discovering conserved coordinates.

### 3.2 HAMILTONIAN NEURAL NETWORK

Hamiltonian Neural Networks (HNNs) (Greydanus et al., 2019) parametrise the Hamiltonian of a conservative dynamical system with a black box MLP $\mathcal{H}_\theta$. The time evolution of coordinates is obtained using Hamilton's equations defined in Equation (1). The Hamiltonian is learned by minimising the distance between the observed change in generalised coordinates and the computed gradients of the learned Hamiltonian

$$\mathcal{L}_{\text{HNN}} = \left\| \frac{\partial \mathcal{H}_\theta}{\partial \mathbf{p}} - \frac{d\mathbf{q}}{dt} \right\|_2 + \left\| \frac{\partial \mathcal{H}_\theta}{\partial \mathbf{q}} + \frac{d\mathbf{p}}{dt} \right\|_2. \tag{3}$$

Because HNNs are trained for single step prediction, and forecasting over long time horizons is performed using Euler integration, numerical errors accumulate and energy drift may be observed even though the inductive bias enforces energy conservation.

To minimize energy drift, Symplectic Recurrent Neural Network (Chen et al., 2019) optimizes over multi-step predictions and integrates forward in time using Leapfrog integration which preserves quadratic invariance. This strategy assumes the Hamiltonian is separable, meaning it can be written as the sum of the potential and kinetic energy $\mathcal{H} = T(p) + V(q)$. In this case, Hamilton's equations can be expressed $\dot{q} = T'(p)$ and $\dot{p} = -V'(q)$, and the symplectic Leapfrog integration is given by

$$p_{n+1/2} = p_n - 1/2 \Delta t T'(q_n), \tag{4}$$

$$q_{n+1} = q_n + \Delta t V'(p_{n+1/2}), \tag{5}$$

$$p_{n+1} = p_{n+1/2} - 1/2 \Delta t T'(q_{n+1}). \tag{6}$$

While the Hamiltonian in our latent motion model (Section 4.2) is modelled by a single network, we assume separability and integrate Hamilton's equations with symplectic Leapfrog integration.

# 4 SYMPLECTIC PHASE SPACE GAN

In this section we present Symplectic Phase Space GAN (SPS-GAN). Given observations of dynamical systems, our goal is to learn the configuration space of the underlying dynamics, which allows us to generate physically consistent trajectories on both the latent and observation space. We achieve this by using a three-stage GAN model conditioned on the physical properties of the dynamical system. SPS-GAN incorporates a configuration space map that maps the latent distribution to an intermediate space we interpret as the motion manifold of the dynamical system. Samples from this space are evolved forward in time using Hamilton's equations and a symplectic Leapfrog integrator. The latent trajectories are decoded into Cartesian trajectories or videos for discrimination. SPS-GAN is built on a generative backbone as it allows us to learn the intrinsic dimension of the configuration space for dynamical systems from data without domain knowledge through our configuration space map (Section 4.1) and sparsity inducing cyclic coordinate loss (Section 4.3).

## 4.1 CONFIGURATION SPACE MAP

The function $f(\epsilon_m, \xi) : \mathbb{R} \to \mathbb{R}^{2 \times d_{lat}}$, conditioned on $\xi = [\text{system label, physical parameters}, \ldots]$, maps a random motion sample $\epsilon_m \sim \mathcal{N}(0, 1)$ onto an intermediate space which we interpret as the configuration space of the underlying dynamical system. As SPS-GAN does not assume a priori knowledge of the dynamical systems, this configuration space has an arbitrary dimension $d_{lat}$.

## 4.2 LATENT SPACE MOTION MODEL

The motion on the latent space is governed by an HNN (Greydanus et al., 2019) conditioned on the system label and physical parameters $\xi$. The initial condition of the system is given by $\mathbf{z}_0 = f(\epsilon_m, \xi)$ where $\epsilon_m$ is a motion sample and $\mathbf{z}_0 = (q_0, p_0)$. We compute a latent trajectory $\{\mathbf{z}_\tau\}_{\tau=0}^T$ of length $T$ by evolving the initial condition forward using the learned Hamiltonian $\mathcal{H}_\theta(q_t, p_t, \xi)$

$$\mathbf{z}_{t+1} = \mathbf{z}_t + \int_t^{t+1} \dot{\mathbf{z}}_\tau \, d\tau = \mathbf{z}_t + \int_t^{t+1} \left( \frac{\partial \mathcal{H}_\theta(q_\tau, p_\tau, \xi)}{\partial p}, -\frac{\partial \mathcal{H}_\theta(q_\tau, p_\tau, \xi)}{\partial q} \right) d\tau. \tag{7}$$

Each motion vector is concatenated with a random content vector $\boldsymbol{\epsilon}_c \in \mathbb{R}^{d_{cont}} \sim \mathcal{N}(\mathbf{0}, \mathbf{I}_{d_{cont}})$ to form the full latent sequence $\{[\mathbf{z}_\tau, \boldsymbol{\epsilon}_c]\}_{\tau=0}^T$. This latent sequence is then passed through the generator to synthesize Cartesian trajectories (Section 4.4) or videos (Section 4.5).

## 4.3 CYCLIC COORDINATE LOSS

As SPS-GAN does not assume a priori knowledge of the generalized coordinates, the learned latent trajectory $\mathbf{z}_t$ is of size $d_{lat}$. To constrain the learned configuration space for interpretability, we leverage the fact that cyclic coordinates do not contribute to the Hamiltonian of the system (see equation (2)) and incorporate a cyclic coordinate loss

$$\mathcal{L}_{cyclic} = \lambda_{cyclic} \sum |\dot{p}_i|, \tag{8}$$

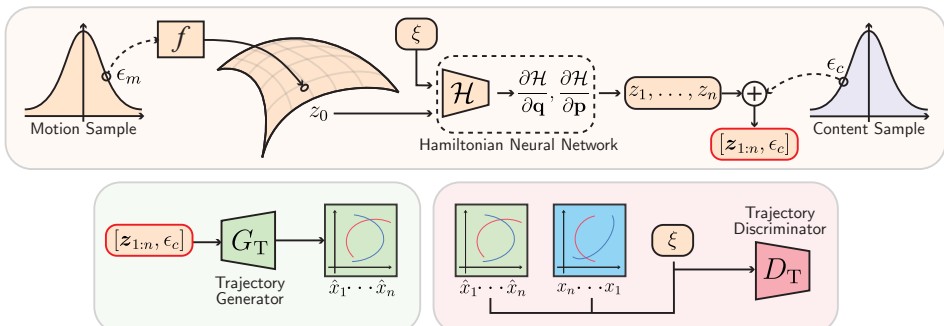

Figure 1: **SPS-GAN-traj for generating Cartesian trajectory.** The random motion sample $\epsilon_m$ is mapped onto the manifold for the dynamical system to create the latent initial condition of the system. The initial conditional is propagated in time using an HNN block to generate a trajectory on the latent space. Cartesian trajectories are generated by passing the latent trajectories through the $G_T$.

where we assign the second half of $\mathbf{z}_t$ to be the momentum terms $p$. This loss encourages the learned configuration space to have a minimal dimension by regularising the momentum terms.

## 4.4 GENERATING CARTESIAN TRAJECTORIES

When generating Cartesian trajectories, SPS-GAN-traj (shown in Figure 1) uses an MLP $G_T$ : $\mathbb{R}^{2\times(d_{\text{lat}}+d_{\text{cont}})} \to \mathbb{R}^{2\times d_{\text{out}}}$ to decode each entry of the latent sequence $\{[\mathbf{z}_\tau, \boldsymbol{\epsilon}_c]\}_{\tau=0}^T$ so that

$$\tilde{\mathbf{x}}_t = G_T([\mathbf{z}_t, \boldsymbol{\epsilon}_c]) \in \mathbb{R}^{2\times d_{\text{out}}} \tag{9}$$

are the coordinates of $d_{\text{out}}$ particles, where $d_{\text{out}}$ is determined by the system in our dataset with the largest number of particles. A system specific binary mask $\mathbf{M}(\xi)$ is applied to the decoded output $\tilde{\mathbf{x}}_t$ so that the entries of inactive particles is 0. The final generated Cartesian trajectory is $\hat{\mathbf{x}} = [\hat{\mathbf{x}}_0, \ldots, \hat{\mathbf{x}}_T]$, where

$$\hat{\mathbf{x}}_t = \tilde{\mathbf{x}}_t \cdot \mathbf{M}(\xi). \tag{10}$$

The binary mask for each system is assumed to be known a priori. This assumption is reasonable since the number of particles for each system is provided in the training dataset. The real trajectory $\mathbf{x}_t \in \mathbb{R}^{2\times d_{\text{out}}}$ is structured similarly to the generated trajectory; 0s are used to pad inactive particles to match the generated outputs for easier computation. The discriminator $D_T$ is a recurrent neural network inspired by the temporal modelling strategy in TimeGAN (Yoon et al., 2019). However, instead of performing discrimination on the latent space, SPS-GAN directly evaluates the generated trajectories. SPS-GAN minimises the binary cross entropy between the generated ($\hat{\mathbf{x}} \sim p_{G_T}$) and real ($\mathbf{x} \sim p_X$) Cartesian trajectories,

$$\mathcal{L}_{\text{traj}} = \mathbb{E}_{\mathbf{x}\sim p_X}\left[\log D_T(\mathbf{x})\right] + \mathbb{E}_{\hat{\mathbf{x}}\sim p_{G_T}}\left[\log\left(1 - D_T(\hat{\mathbf{x}})\right)\right] + \mathcal{L}_{\text{cyclic}}. \tag{11}$$

## 4.5 GENERATING VIDEOS

When generating video data, SPS-GAN-video (shown in Figure 2) leverages a CNN decoder $G_I$ : $\mathbb{R}^{2\times d_{\text{lat}}+d_{\text{cont}}} \to \mathbb{R}^{d_{\text{h}}\times d_{\text{w}}\times 3}$ to decode each entry of the latent sequence $\{[\mathbf{z}_\tau, \boldsymbol{\epsilon}_c]\}_{\tau=0}^T$ into RGB images. The generated video $\hat{\mathbf{v}} = [\hat{\mathbf{x}}_0, \ldots \hat{\mathbf{x}}_T]$ is the concatenation of a sequence of images, each generated by $G_I$ where

$$\hat{\mathbf{x}}_t = G_I([\mathbf{z}_t, \boldsymbol{\epsilon}_c]) \in \mathbb{R}^{d_{\text{h}}\times d_{\text{w}}\times 3}. \tag{12}$$

The image discriminator $D_I$ takes generated and real images as input and is designed to enforce per-frame realism of the generated images. The video discriminator $D_V$ takes generated and real videos as input and is designed to enforce dynamical coherence. Together, the discriminators ensure realistic content (size of particles, colour of particles, etc.) and realistic dynamics. SPS-GAN minimises the binary cross entropy between the generated ($\hat{\mathbf{v}} \sim p_{G_I}$) and real ($\mathbf{v} \sim p_V$) videos,

$$\mathcal{L}_{\text{video}} = \mathbb{E}_{\mathbf{v}\sim p_V}\left[\log D_I(\mathbf{x}_t) + \log D_v(\mathbf{v})\right] \tag{13}$$
$$+ \mathbb{E}_{\hat{\mathbf{v}}\sim p_{G_I}}\left[\log\left(1 - D_I(\hat{\mathbf{x}}_t)\right) + \log\left(1 - D_v(\hat{\mathbf{v}})\right)\right] + \mathcal{L}_{\text{cyclic}}.$$

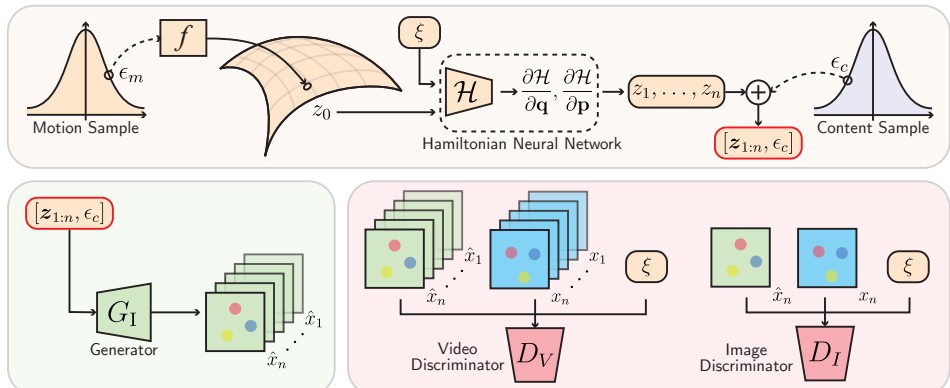

Figure 2: **SPS-GAN-vid for generating video.** The random motion sample $\epsilon_m$ is mapped onto the manifold for the dynamical system to create the initial condition of the system. The initial conditional is propagated in time using a HNN block to generate the trajectory on the latent space, which is concatenated with content sample $\epsilon_c$. Video frames are generated by passing the latent trajectories through $G_I$. Discriminators $D_V$ and $D_I$ ensures realistic dynamics and content respectively.

## 5 EXPERIMENT

This section highlights the ability of SPS-GAN to generate accurate and consistent trajectories, model multiple systems, and discover system symmetries.

### 5.1 GENERATING CARTESIAN TRAJECTORIES AND DISCOVERING SYMMETRIES

We evaluate SPS-GAN using the Cartesian trajectories of the mass-spring oscillator, ideal pendulum, double pendulum, planar two-body systems, and planar three-body systems. Since these dynamical systems can be accurately simulated, and some of them (e.g., double pendulum) exhibit chaotic behaviour, they provide a suitable benchmark for assessing our model's ability to capture both simple and complex dynamics. We compare SPS-GAN against the supervised baseline Hamiltonian Neural Network (HNN) (Greydanus et al., 2019). SPS-GAN, HNN, NeuralODE (Chen et al., 2018), and LatentODE (Rubanova et al., 2019) are trained on the same set of real Cartesian trajectories. At test time, the reference real trajectory is simulated with known dynamics using the first generated Cartesian coordinate from SPS-GAN $\hat{x}_0$ as initial condition. HNN and NeuralODE is then asked to predict the same reference real trajectory using the first coordinate as initial condition. We compare the MSE between the generated/predicted trajectories and the ground truth in Table 1. Our generative SPS-GAN performs on par with supervised HNN. Comparison of the generated Cartesian trajectories are shown in Figure 3 where the generated trajectory closely matches ground truth. This indicates that the configuration space map and latent motion model is able to accurately capture the dynamics and motion manifold of systems in a unsupervised manner. Furthermore, for an ideal pendulum, the generated trajectory has total energy $3.110 \pm 0.119$, where the variance is on par with the trajectory predicted by HNN, which has total energy $3.119 \pm 0.110$.

We further evaluate SPS-GAN on generating all five systems simultaneously, with explicit system labels and physical parameters used as the conditioning inputs. The generated Cartesian trajectories

Table 1: **Accuracy of predicted trajectory when modelling a single system.** Comparison of predicted trajectory across mass–spring oscillator, ideal pendulum, double pendulum, planar two-body systems, and planar three-body systems is reported. We report the MSE between predicted and ground truth trajectories of 30 timesteps, with $\Delta t = 0.05$. When generating dynamics from one system at a time, SPS-GAN exhibits predictive accuracy on par with supervised HNN.

|  | Mass-Spring | Pendulum | Double Pendulum | Two-Body | Three-Body |
|---|---|---|---|---|---|
| SPS-GAN | $6.16 \times 10^{-4}$ | $1.91 \times 10^{-3}$ | $\mathbf{3.25 \times 10^{-2}}$ | $3.68 \times 10^{-3}$ | $\mathbf{1.26 \times 10^{-3}}$ |
| HNN | $3.53 \times 10^{-4}$ | $\mathbf{6.27 \times 10^{-4}}$ | $1.25 \times 10^{-1}$ | $\mathbf{3.61 \times 10^{-3}}$ | $2.81 \times 10^{-3}$ |
| NeuralODE | $\mathbf{1.62 \times 10^{-4}}$ | $1.59 \times 10^{-3}$ | $5.99 \times 10^{-2}$ | $5.72 \times 10^{-3}$ | $3.67 \times 10^{-3}$ |
| LatentODE | - | - | $4.66 \times 10^{-2}$ | $1.98 \times 10^{-2}$ | $1.37 \times 10^{-3}$ |

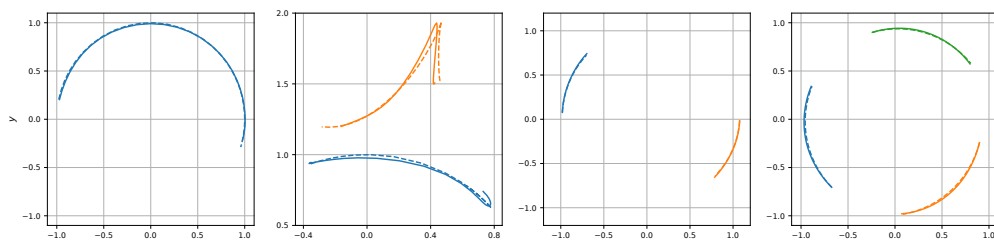

Figure 3: **Accuracy of predicted trajectory when modelling a single system.** Qualitative comparison between ground-truth (dotted) and generated (solid) trajectories across ideal pendulum, double pendulum, planar two-body systems, and planar three-body systems is shown. When generating dynamics from one system at a time, the generated trajectory from SPS-GAN closely follows reference dynamics.

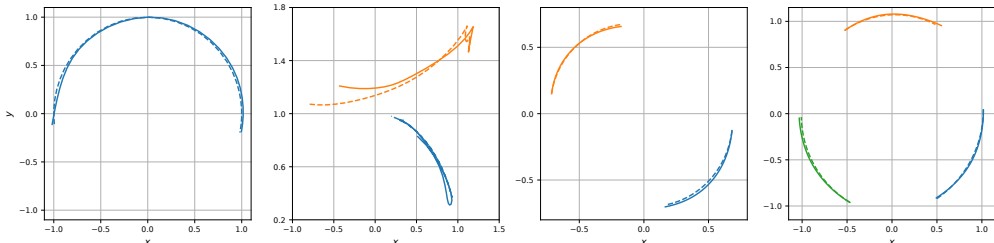

Figure 4: **Accuracy of predicted trajectory when modelling multiple systems.** Qualitative comparison between ground-truth (dotted) and generated (solid) trajectories across ideal pendulum, double pendulum, planar two-body systems, and planar three-body systems is shown. When generating dynamics from five distinct systems SPS-GAN is able to disentangle different dynamics and the generated trajectory from SPS-GAN closely follows reference dynamics.

are compared with the ground truth in Figure 4, where we show SPS-GAN can disentangle the different dynamics and generate accurate trajectories for each system, demonstrating an ability to represent multiple dynamics in a single model.

A core contribution of SPS-GAN is the ability to identify symmetries by minimising the dimension of the latent space. To understand the structure of the learned latent space, we project the latent motion trajectories using t-SNE in Figure 5. In all experiments, we set the dimension of the latent space $d_{\text{lat}}$ to be 20. The projection shows that SPS-GAN is able to correctly identify a the two-body system as 1-dimensional, and the double pendulum and three-body systems as 2-dimensional. Using the same data, FastICA yields results that have incorrectly structured latent spaces.

**Double pendulum.** The double pendulum is an example of a chaotic Hamiltonian system with two angular degrees of freedom. The system Hamiltonian given by

$$\mathcal{H}\left(q_1, q_2, p_1, p_2\right) = \frac{1}{2mL^2} \cdot \frac{p_1^2 + 2p_2^2 - 2p_1 p_2 \cos(q_1 - q_2)}{1 + \sin^2(q_1 - q_2)} + mgL\left(3 - 2\cos q_1 - \cos q_2\right).$$

Because both generalised coordinates are independent and coupled nonlinearly through the $\cos\left(q_1 - q_2\right)$ term, the system requires at least two generalised coordinates to represent its dynamics. While SPS-GAN generates Cartesian trajectories that are naively represented with 4 degrees of freedom, $(x_1, y_1, x_2, y_2)$, it is able to correctly identify that the minimal degrees of freedom required to represent the system is two (see Figure 5).

**Two-body.** The Newtonian two-body problem (masses $m_1, m_2$ at positions $\mathbf{r}_1, \mathbf{r}_2$) is governed by the motion of its center of mass by the radial equations

$$\dot{r}^2 = \frac{2}{\mu}(E - V_{\text{eff}}(r)), \qquad V_{\text{eff}}(r) = \frac{L^2}{2\mu r^2} + V(r). \tag{14}$$

For trajectories with conserved energy $E$ and angular momentum $L$, the radial motion lies on a 1-dimensional manifold parametrised by $r$ (a detailed derivation is given in Appendix B.2). While SPS-GAN generates Cartesian trajectories that are naively represented with 4 degrees of freedom

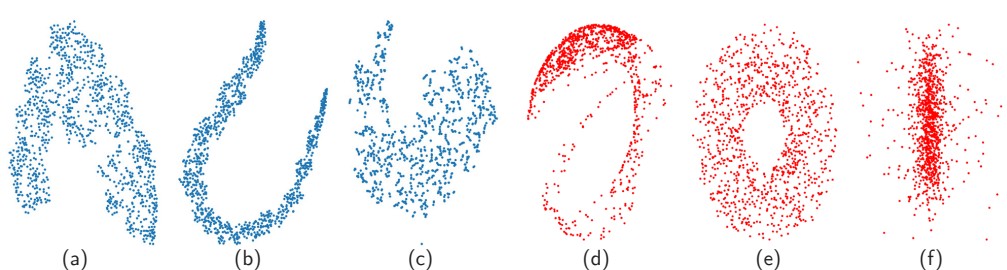

(a)      (b)      (c)      (d)      (e)      (f)

Figure 5: **T-SNE projection of learned latent space. (a-c)** t-SNE projection of learned latent spaces for the double pendulum, planar two-body, and planar three-body; **(d-e)** FastICA (Hyvärinen & Oja, 2000) projection of trajectories for the double pendulum, planar two-body, and planar three-body. The t-SNE projections indicate that the learned latent dimensions is 1 for two-body and 2 for double pendulum and three-body while FastICA on the trajectory shows no discernable structure. This matches physical intuition for the three systems and indicates that SPS-GAN is able to learned the correctly sized latent dimension without supervision and a priori knowledge.

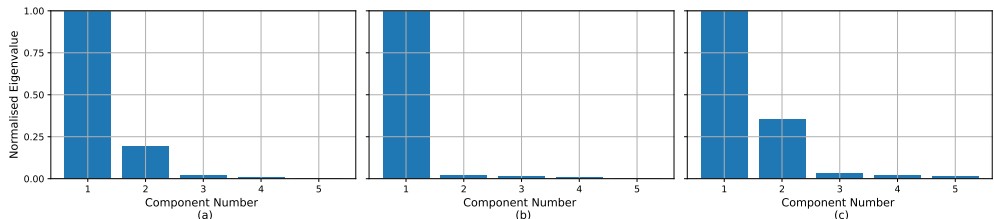

Figure 6: **PCA eigenvalue analysis of the learned latent space. (a-c)** The first 5 principal components for the learned latent space for double pendulum, planar two-body, and planar three-body. The eigenvalues are normalised with respect to the largest component. The Cattell's criterion indicate that the learned latent dimension is 1 for two-body and 2 for double pendulum and three-body. Additionally, the cumulative variance is provided in Table 5 in Appendix B.5.

$(x_1, y_1, x_2, y_2)$, it is able to correctly identify that the minimal degrees of freedom required to represent the system is one (see Figure 5).

**Three body.** The planar three-body problem is governed by Newton's law of gravitation

$$m_i \ddot{q}_i = G \sum_{j \neq i} \frac{m_i m_j (q_j - q_i)}{\|q_j - q_i\|^3}, \quad i = 1, 2, 3. \tag{15}$$

We consider a special case of three-body motion, where all masses are initialised on the vertices of an equilateral triangle. In this configuration, the system evolves in a rotating equilateral triangular with the two degrees of freedom (i.e., the angle of the centre of mass and the distance between bodies). A detailed derivation of the system Hamiltonian and reduction is given in Appendix B.2. While SPS-GAN generates Cartesian trajectories that are naively represented with 6 degrees of freedom $(x_1, y_1, x_2, y_2, x_3, y_3)$, it is able to correctly identify that the minimal degrees of freedom required to represent the system is two (see Figure 5).

## 5.2 GENERATING CARTESIAN TRAJECTORIES OF REAL WORLD SYSTEM

We evaluate SPS-GAN using real world single pendulum trajectory data recorded by Schmidt & Lipson (2009). Comparison of the generated and ground truth Cartesian trajectories are shown in Figure 7 where the generated trajectory closely matches ground truth. The MSE between ground truth and generated is $1.62 \times 10^{-3}$, similar to the error reported for the simulated pendulum in Table 1. Additionally SPS-GAN's learned latent space exhibits approximately 1-dimension, as shown by the eigenvalue analysis in Figure 7. We attribute the small second eigenvalue to the presence of friction.

## 5.3 GENERATING CARTESIAN TRAJECTORIES OF SYSTEMS WITH BIFURCATION

We evaluate SPS-GAN using systems that exhibit bifurcations, whose behaviour is characterized by a sudden appearance of a qualitatively different solution when some system parameter is varied.

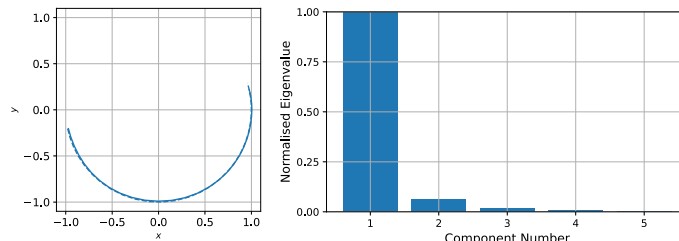

Figure 7: **Learned trajectory and latent space of real pendulum. (Left)** Qualitative comparison between ground-truth and generated trajectory of a real pendulum (Schmidt & Lipson, 2009) shows that the generated trajectory from SPS-GAN closely follows reference dynamics, even for real world systems; (**Right**) PCA eigenvalue analysis for the learned latent space shows approximately a 1-dimensional subspace. The eigenvalues are normalised with respect to the largest component. The small second principal component can be attributed to the non-Hamiltonian contribution of friction.

Specifically, we consider a 1-DoF Hamiltonian saddle-node bifurcation (Lyu et al., 2021) define by

$$\mathcal{H}\left(q,p\right) = \frac{1}{2}p^2 - \mu x + \frac{\alpha}{3}x^3. \tag{16}$$

SPS-GAN is trained on the regime with $\mu < 0$ and the learned model is tested on the the regime with $\mu > 0$. We assume that the 2-dimension Cartesian coordinates of the resulting trajectories to be of the form $(x, y) = (q, 0)$. The MSE between generated and ground truth trajectory when $\mu = -1$ is $2.72 \times 10^{-3}$, and when $\mu = 1$ is $8.83 \times 10^{-3}$. While there is a slight reduction in performance, the post-bifurcation accuracy remains the same order of magnitude as pre-bifurcation.

### 5.4 GENERATING VIDEOS OF SINGLE SYSTEMS

We evaluate SPS-GAN when it is tasked with modelling a single system with constant physical parameters. The results are baselined against HGN (Toth et al., 2019), which learns a latent representation of the phase-space of a single system in a VAE, and evolves latent samples forward in time with a learned HNN, and Hamiltonian GAN (Allen-Blanchette, 2024), learns to generate conservative trajectories using a learned HNN in a GAN framework. This work differs from ours in that it can only capture the dynamics of a single system. The Fréchet Video Distance (FVD) metric (Unterthiner et al., 2018), computed over 2048 generated videos of 16 frames long, for the mass-spring oscillator, ideal pendulum, double pendulum, two-body system, and three-body system is reported in Table 2. Sample videos generated by SPS-GAN are shown in Figure 10 in Appendix B.3. SPS-GAN significantly outperforms all baselines across all five physical systems. Notably, SPS-GAN reduces FVD by more than an order of magnitude compared to HGN on the chaotic double pendulum system. SPS-GAN also exhibited improved performance compared to HGAN, which implements a similar cyclic coordinate loss, due to the conditioning on system labels.

Table 2: **Consistency of generated video when modelling a single system.** FVD of generated video across mass–spring oscillator, ideal pendulum, double pendulum, planar two-body systems, and planar three-body systems is reported for HGAN (Allen-Blanchette, 2024), and HGN (Toth et al., 2019). When generating video from one system at a time, SPS-GAN exhibits consistency significantly higher than baseline models.

| | Mass-Spring | Pendulum | Double Pendulum | Two-Body | Three-Body |
|---|---|---|---|---|---|
| SPS-GAN | **25.63** | **40.57** | **24.12** | **87.12** | **89.08** |
| HGAN | 45.68 | 91.64 | 73.21 | 105.85 | 1981.10 |
| HGN | 385.08 | 688.12 | 331.94 | 830.91 | 451.40 |

Table 3: **Consistency of generated video when modelling multiple systems under parameter variation.** FVD of generated video modelling multiple systems is reported. When generating video from multiple systems with variable parameter, SPS-GAN exhibits high physical consistency.

| System | Colour | Physics | FVD |
|---|---|---|---|
| Varied | Varied | Varied | 182.15 |
| Varied | Constant | Varied | 135.38 |
| Varied | Varied | Constant | 135.28 |

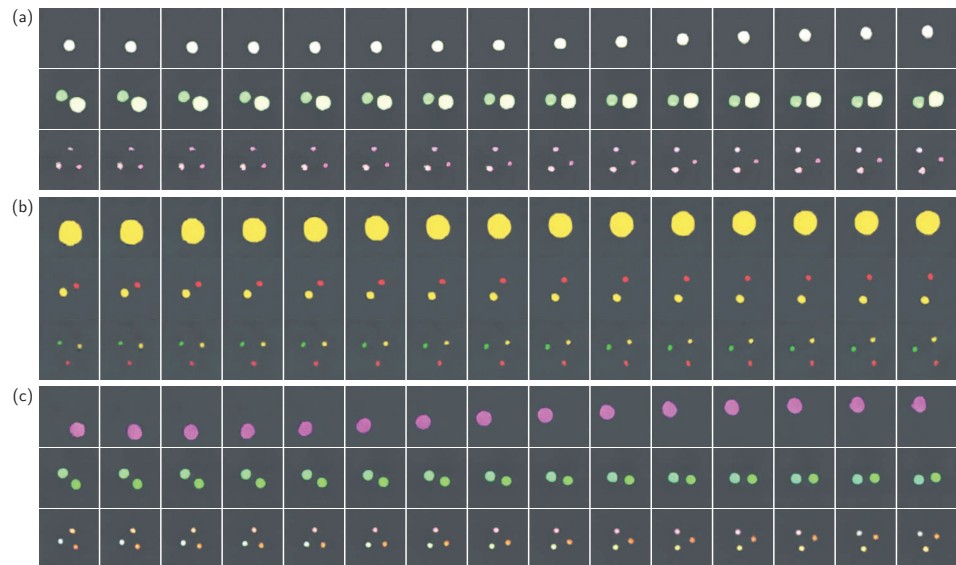

Figure 8: **Consistency of generated video when modelling multiple systems under parameter variation.** Sample generated video of mass-spring oscillator, two-body system, and three-body system under **(a)** physics and colour varied, **(b)** only physics varied, and **(c)** only colour varied.

## 5.5 GENERATING VIDEOS OF MULTIPLE SYSTEMS UNDER PARAMETER VARIATION

We evaluate SPS-GAN when it is tasked with modelling multiple systems with parameter variation that affects both the physical and visual characteristics. Visual variability is introduced through variable colour and physical variability is introduced through altering physical parameters such as mass. This experiment assess the ability of SPS-GAN to generalize across heterogeneous systems and unseen parameters, and disentangle motion from appearance. Table 3 reports the FVD when both physics and colour are varied, when only colour is varied, and when only physics is varied. Notably, in all cases SPS-GAN achieves FVD values lower than those achieved by HGN on constant systems. The model's stable results across all conditions indicate that the conditional Hamiltonian structure is able to disentangled content and motion, enabling robust video generation under diverse sources of variability. Figure 8 showcases sample generated videos under all three settings for the mass-spring oscillator, two-body system, and three-body system. This showcases SPS-GAN's ability to extend the learned motion model to unseen system configurations and parameters.

## 6 CONCLUSION

In this paper we propose SPS-GAN, a general framework to generate both Cartesian trajectory and video data of multiples systems under variable parameters and discover symmetry in dynamical systems. By leveraging the observation that cyclic-coordinates provide a principled way to reduce the learned configuration space, we discover a latent configuration space that is able to produce physically plausible trajectories as well as a identify cyclic-coordinates. SPS-GAN demonstrates its ability to generate trajectory data with accuracy on par with supervised baselines and video data under different settings with consistency significantly higher than existing physics-informed baselines.

**Future Work.** SPS-GAN relies on a standard RNN based discriminator. Existing work have used hand picked Hamiltonians as both the genrator and discriminator for quantum applications (Kim et al., 2024). Future work will investigate embedding general Hamiltonian structure in the discriminator.

**Limitations.** While the paper demonstrates the utility of SPS-GAN, we do not extend to use the learned representations for additional tasks. Future work will focus on applying the proposed method on unknown systems to discover hidden symmetries that can be used in downstream tasks such as energy based control. While the latent space is governed by Hamiltonian dynamics, the decoding MLP is unconstrained, leading to output trajectories that do not strictly follow conservation of energy in the Cartesian or video representation.

## ETHICS STATEMENT

While the authors do not see a direct way that SPS-GAN can be used to cause harm for both the ML community and broader society, we acknowledge that the core contribution of generating physically plausible videos can be used to create generated content that appear more consistent for malicious intent, such as spreading misinformation and enforcing societal stereotypes. SPS-GAN is trained on publicly available synthetic datasets of dynamical systems, that have been sufficienctly cited within text, with no human subjects.

## REPRODUCIBILITY STATEMENT

The SPS-GAN code will be made publicly available upon acceptance of the paper. All hyperparameters required for running experiments presented in Section 5 are given in Appendix D, along with hardware requirements. All video experiments in this paper are trained on datasets produced from Rodas et al. (2020) and Schmidt & Lipson (2009), with model parameters given in Appendix A. The generation of training time-series data is modified from the video generation pipeline to output low-dimensional measurement data. This pipeline will be made public along with the repo upon acceptance.

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

## A   DATASET

In this section we include additional details on the training dataset used to train SPS-GAN and baseline models on experiments in Section 5. We use the data generation pipeline from the publicly availible re-implementation of HGN (Rodas et al., 2020). The dataset consists of five dynamical systems, including mass-spring oscillator, ideal pendulum, double pendulum, two-body problem, and three-body problem. After sampling the initial conditions, all systems were simulated using the RK45 with a time step of $\Delta t = 0.05$, producing trajectories of 30 frames each.

**Mass-Spring System.** The damped harmonic oscillator is represented by the Hamiltonian

$$H = \frac{p^2}{2m} + \frac{1}{2}kq^2, \tag{17}$$

where the dynamics are governed by

$$\ddot{q} = -2c\sqrt{k/m}\dot{q} - (k/m)q, \tag{18}$$

with mass $m = 0.5$, elastic constant $k = 2.0$, and zero damping. Initial conditions are sampled within radius bounds $[0.1, 1.0]$.

**Ideal Pendulum.** The ideal pendulum system is represented by the Hamiltonian

$$H = \frac{p^2}{2mL^2} + mgL(1 - \cos\theta), \tag{19}$$

where the dynamics are governed by

$$\ddot{\theta} = -(g/L)\sin\theta, \tag{20}$$

with mass $m = 0.5$, length $L = 1.0$, and gravity $g = 3.0$. Initial conditions are sampled within radius bounds $[1.3, 2.3]$.

**Double Pendulum.** The chaotic double pendulum system is represented by the Hamiltonian

$$H = \frac{1}{2mL^2} \cdot \frac{p_1^2 + 2p_2^2 - 2p_1p_2\cos(\theta_1 - \theta_2)}{1 + \sin^2(\theta_1 - \theta_2)} + mgL(3 - 2\cos\theta_1 - \cos\theta_2) \tag{21}$$

with parameters $m = 1.0$, $L = 1.0$, and $g = 3.0$. Initial conditions of each object is sampled within radius bounds $[0.5, 1.3]$.

**Two-Body System.** The gravitational two-body system is represented by the Hamiltonian

$$H = \frac{|\mathbf{p}_1|^2}{2m_1} + \frac{|\mathbf{p}_2|^2}{2m_2} - \frac{Gm_1m_2}{|\mathbf{q}_1 - \mathbf{q}_2|}, \tag{22}$$

where the dynamics are governed by

$$m_i\ddot{\mathbf{q}}_i = G\sum_{j \neq i} \frac{m_im_j(\mathbf{q}_j - \mathbf{q}_i)}{|\mathbf{q}_j - \mathbf{q}_i|^3}, \tag{23}$$

with equal masses $m = 1.0$ and gravitational constant $G = 1.0$. The two objects are placed on the same circular orbit at opposite position with the radius ranging from 0.5 to 1.5. Their initial velocities are set perpendicular to the radius vector, and a small perturbations of magnitude 0.1 is added to the velocities.

**Three-Body System.** The gravitational three-body system is represented by the Hamiltonian

$$H = \sum_{i=1}^{3} \frac{|\mathbf{p}_i|^2}{2m_i} - \sum_{i<j} \frac{Gm_im_j}{|\mathbf{q}_i - \mathbf{q}_j|}, \tag{24}$$

with equal masses $m = 1.0$ and gravitational constant $G = 1.0$. Three equal mass objects are initialized in an equilateral triangle configuration with $120°$ angular separation within radius bounds $[0.9, 1.2]$. Their initial velocities are set perpendicular to the radius vector, and a small perturbations of magnitude 0.1 is added to the velocities.

## B  ADDITIONAL RESULTS

In this appendix section we include additional details to the experiments presented in the main body in this paper, including detailed derivation of Hamiltonians and degrees of freedom, and additional sampled trajectories.

### B.1  CONSERVATION OF ENERGY IN GENERATED TRAJECTORIES

We report the energy drift between the generated time-series data and the ground truth solution in Figure 9. As SPS-GAN does not output momentum, we use central difference to calculate the momentum of the pendulum bob to calculate the kinetic and potential energy. The parameters for the ideal pendulum system is given in Appendix A.

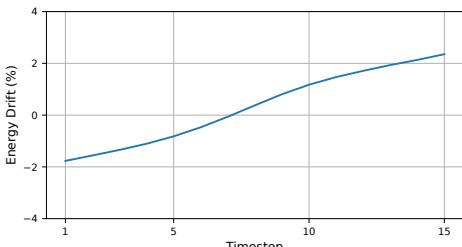

Figure 9: **Conservation of energy in generated trajectories.** Energy drift in percentages with respect to the ground truth constant energy for the ideal pendulum system is reported. The generated trajectories exhibits energy variance within 2.5% of the ground truth values.

### B.2  DERIVATION OF SYSTEM HAMILTONIAN AND DEGREE OF FREEDOMS

Constraints and structure provides a structured and interpretable way to reduce the dimensionality of the system being modelled (Kasim & Lim, 2022; Matsubara & Yaguchi, 2022; Yang et al., 2024a). By leveraging cyclic coordinates we are able to discover the dimensionality of the underlying dynamical structure for the three systems presented in Section 5.1. We expand on the discussion given in the main body to show the analytical form of the degrees of freedom SPS-GAN was able to discover.

**Two Body Problem.** SPS-GAN is given observations of the two body system in terms of positions $\mathbf{r}_1, \mathbf{r}_2$. For a Newtonian two-body problem (masses $m_1, m_2$ at positions $\mathbf{r}_1, \mathbf{r}_2$), its dynamics is governed by equations of motion

$$m_1 \ddot{\mathbf{r}}_1 = -G \frac{m_1 m_2 (\mathbf{r}_1 - \mathbf{r}_2)}{\|\mathbf{r}_1 - \mathbf{r}_2\|^3}, \tag{25}$$

$$m_2 \ddot{\mathbf{r}}_2 = -G \frac{m_1 m_2 (\mathbf{r}_2 - \mathbf{r}_1)}{\|\mathbf{r}_2 - \mathbf{r}_1\|^3}. \tag{26}$$

By introducing the center of mass

$$\mathbf{R} = \frac{m_1 \mathbf{r}_1 + m_2 \mathbf{r}_2}{M}, \tag{27}$$

where $M = m_1 + m_2$, and the relative position vector between the two masses

$$\mathbf{r} = \mathbf{r}_1 - \mathbf{r}_2, \tag{28}$$

one can obtain the reduced equation for the relative motion of the reduced mass $\mu = (m_1 m_2)/M$

$$\mu \ddot{\mathbf{r}} = -G \frac{m_1 m_2}{\|\mathbf{r}\|^3} \mathbf{r} \implies \ddot{\mathbf{r}} = -G \frac{M}{\|\mathbf{r}\|^3} \mathbf{r}. \tag{29}$$

Moving to polar coordinates $(r, \theta)$ for the relative vector $\mathbf{r}$, and leveraging conservation of angular momentum $L = \mu r^2 \dot{\theta}$, the conserved energy of the system can be defined as

$$E = \frac{1}{2} \mu \dot{r}^2 + \frac{L^2}{2\mu r^2} + V(r), \qquad V(r) = -G \frac{m_1 m_2}{r}. \tag{30}$$

Rearranging (30) for the expression of $\dot{r}$ yields

$$\dot{r}^2 = \frac{2}{\mu}(E - V_{\text{eff}}(r)), \qquad V_{\text{eff}}(r) = \frac{L^2}{2\mu r^2} + V(r). \tag{31}$$

Therefore, for trajectories with conserved energy $E$ and angular momentum $L$, the radial motion lies on an one-dimensional manifold parametrised by $r$. This derivation matches the dimension of the t-SNE projection for the learned motion manifold of the two body system shown in Figure 5.

**Three body problem.** We denote by $q_j \in \mathbb{R}^2$ the position vector of the $j$th particle of mass $m_j, j = 1, 2, 3$. Define the collision set as

$$\Delta_{ij} = \{q = (q_1, q_2, q_3) \in \mathbb{R}^6 | q_i = q_j\} \quad \Delta = \cup_{1 \le i < j \le 3} \Delta_{ij}. \tag{32}$$

We consider $q = (q_1, q_2, q_3) \in \mathbb{R}^6 \setminus \Delta$ and the mass matrix $M = \text{diag}(m_1, m_1, m_2, m_2, m_3, m_3)$. The total potential and Newton's equations of motion are given by

$$U(q) = \sum_{1 \le i < j \le 3} \frac{m_i m_j}{\|q_i - q_j\|}, \qquad M\ddot{q} = \nabla U(q) \tag{33}$$

We seek the homographic solutions of the form

$$q_j(t) = z(t)\, q_j^{(c)}, \qquad j = 1, 2, 3, \tag{34}$$

where $q^{(c)} = (q_1^{(c)}, q_2^{(c)}, q_3^{(c)}) \in \mathbb{R}^6 \setminus \Delta$ is fixed and $z(t) \in \mathbb{C}$ controls global scale and rotation. Plugging the provided form into Newton's Equations (33), we have

$$\ddot{q}_j = \ddot{z}(t)q_j^{(c)}, \quad \nabla U(q(t)) = \|z(t)\|^{-3} z(t)\nabla U(q^{(c)}), \tag{35}$$

$$\ddot{z}(t)Mq^{(c)} = \|z(t)\|^{-3} z(t)\nabla U(q^{(c)}), \tag{36}$$

which implies

$$\ddot{z}(t) = -\lambda \frac{z(t)}{|z(t)|^3}, \tag{37}$$

where $\lambda$ is determined by

$$-\lambda M q^{(c)} = \nabla U(q^{(c)}). \tag{38}$$

Let $z(t) = r(t)e^{i\phi(t)}$ and take $\lambda = 1$, then

$$\ddot{z} = (\ddot{r} - r\dot{\phi}^2)e^{i\phi} + (2\dot{r}\dot{\phi} + r\ddot{\phi})ie^{i\phi}. \tag{39}$$

Equating real and imaginary parts, we have

$$r\ddot{\phi} + 2\dot{r}\dot{\phi} = 0, \quad \ddot{r} - r\dot{\phi}^2 = -\frac{1}{r^2}, \tag{40}$$

and applying conservation of angular momentum

$$L = \sum_{i=1}^{3} m_i \|q_i\|^2 r^2 \dot{\phi} \quad \omega = r^2 \dot{\phi} = C, \tag{41}$$

yield the single second-order ODE

$$\ddot{r} = -\frac{1}{r^2} + \frac{\omega^2}{r^3}. \tag{42}$$

Thus the Newtonian two-body problem collapses to 2 DOF under homographic constrains. This provides a plausible explanation for the 2D t-SNE embedding observed for the planar three-body dataset. Similarly, one can also identify the two degree of freedom by representing the solutions to the three body problem by viewing the configurations of triangles on the shape sphere (Montgomery, 2015; Kendall, 1984).

### B.3 GENERATING CONSTANT SETTING VIDEO DATA

We include sample trajectories of SPS-GAN-vid when modelling single systems under constant parameter setting in Figure 10.

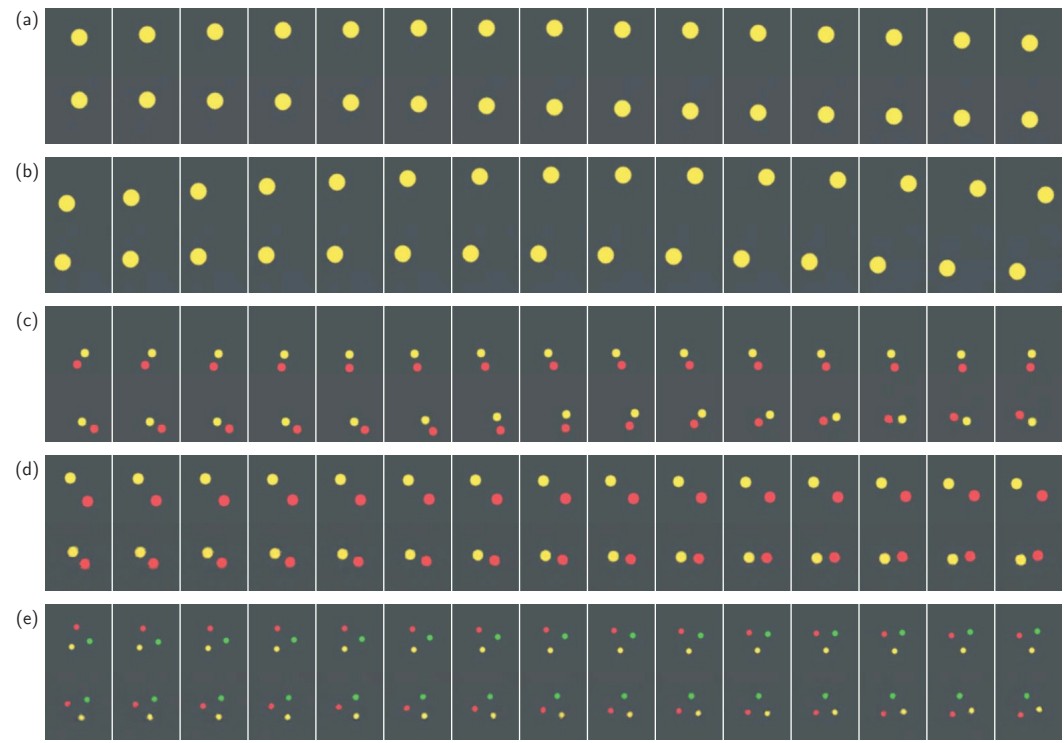

Figure 10: **Consistency of generated video when modelling a single system under constant parameter.** Real system (top rows) and sample generated video (bottom row) of **(a)** mass-spring oscillator, **(b)** ideal pendulum, **(c)** double pendulum, **(d)** two body, and **(e)** three body system under constant parameter.

Table 4: **Accuracy of predicted trajectory at different rollout lengths.** Comparison of the learned trajectory across (**Top-to-Bottom**) mass-spring, ideal pendulum, double pendulum, planar two-body systems, and planar three-body systems at rollout lengths $[5, 10, 20, 25, 30]$ with $\Delta t = 0.05$ is reported. When generating single systems, SPS-GAN exhibits predictive performance on par with supervised HNN at a different rollout lengths.

| Rollout | 5 | 10 | 20 | 25 | 30 |
|---|---|---|---|---|---|
| SPS-GAN | $2.69 \times 10^{-5}$ | $1.91 \times 10^{-4}$ | $6.13 \times 10^{-4}$ | $6.65 \times 10^{-4}$ | $6.16 \times 10^{-4}$ |
| HNN | $7.19 \times 10^{-6}$ | $3.92 \times 10^{-5}$ | $1.83 \times 10^{-4}$ | $2.67 \times 10^{-4}$ | $3.53 \times 10^{-4}$ |
| Rollout | 5 | 10 | 20 | 25 | 30 |
| SPS-GAN | $4.01 \times 10^{-5}$ | $1.55 \times 10^{-4}$ | $6.61 \times 10^{-4}$ | $1.15 \times 10^{-3}$ | $1.91 \times 10^{-3}$ |
| HNN | $2.85 \times 10^{-5}$ | $1.23 \times 10^{-4}$ | $3.96 \times 10^{-4}$ | $5.09 \times 10^{-4}$ | $6.27 \times 10^{-4}$ |
| Rollout | 5 | 10 | 20 | 25 | 30 |
| SPS-GAN | $1.89 \times 10^{-3}$ | $3.22 \times 10^{-3}$ | $1.78 \times 10^{-2}$ | $2.45 \times 10^{-2}$ | $3.25 \times 10^{-2}$ |
| HNN | $1.76 \times 10^{-3}$ | $1.10 \times 10^{-2}$ | $7.35 \times 10^{-2}$ | $8.57 \times 10^{-2}$ | $1.25 \times 10^{-1}$ |
| Rollout | 5 | 10 | 20 | 25 | 30 |
| SPS-GAN | $5.25 \times 10^{-6}$ | $1.02 \times 10^{-4}$ | $1.29 \times 10^{-3}$ | $2.81 \times 10^{-3}$ | $3.68 \times 10^{-3}$ |
| HNN | $9.51 \times 10^{-6}$ | $4.31 \times 10^{-4}$ | $1.81 \times 10^{-3}$ | $2.99 \times 10^{-3}$ | $3.61 \times 10^{-3}$ |
| Rollout | 5 | 10 | 20 | 25 | 30 |
| SPS-GAN | $2.07 \times 10^{-6}$ | $2.13 \times 10^{-5}$ | $2.78 \times 10^{-4}$ | $6.42 \times 10^{-4}$ | $1.26 \times 10^{-3}$ |
| HNN | $4.86 \times 10^{-5}$ | $2.37 \times 10^{-4}$ | $1.10 \times 10^{-3}$ | $1.83 \times 10^{-3}$ | $2.81 \times 10^{-3}$ |

## B.4 TRAJECTORY PREDICTION AT DIFFERENT ROLL-OUT LENGTHS

We compared the MSE error between the predicted trajectory and ground truth of models trained at different roll-out lengths $[5, 10, 20, 25, 30]$. We report the results in Table 4 where in general SPS-GAN performs on par with supervised HNN.

## B.5 PCA EIGENVALUE ANALYSIS

In Section 5, the number of principal components is determined using Cattell criterion on the Scree plot given in Figure 6. However, some have criticised the Cattell criterion as being subjective (Raîche et al., 2013). To ensure our analysis is more robust, we also report the first 5 cumulative variance from the PCA projection in Table 5, and use the canonical rule of $\sim 95\%$ as the cut off to determine the number of dominant dimensions (Abdi & Williams, 2010).

Table 5: **Cumulative variance of PCA eigenvalue analysis.** The cumulative variance of the first 5 principal components is reported for the double pendulum, two-body, and three-body systems.

| System | 1 | 2 | 3 | 4 | 5 |
|---|---|---|---|---|---|
| Double Pendulum | 76.98 | 96.82 | 97.73 | 98.38 | 98.90 |
| Two-Body | 94.05 | 95.95 | 97.39 | 98.12 | 98.63 |
| Three-Body | 76.64 | 95.25 | 97.95 | 98.77 | 99.18 |

## C ABLATION STUDY

In this appendix section we include ablation study on SPS-GAN. This section studies the effect of the cyclic coordinate loss (Appendix C.1), HNN module (Appendix C.2), and Hamiltonian assumption (Appendix C.3) on the performance of the model.

## C.1 ABLATION STUDY ON CYCLIC COORDINATE LOSS

We perform ablation study of the cyclic coordinate loss for the two-body system. In Section 5, SPS-GAN was able to identify a 1-dimension latent space, as shown in Figure 5 and 6. By reducing the weighting of the cyclic coordinate loss by $1/3$ and then setting it to $0$, we see that the second eigenvalue grows in magnitude, indicating SPS-GAN is now learning a 2-dimensional representation. This goes against the analytical anaylysis given in Appendix B.2. The weight for the cyclic coordinate loss can be determined by sweeping across values and choosing the smallest values that gives the minimal representation.

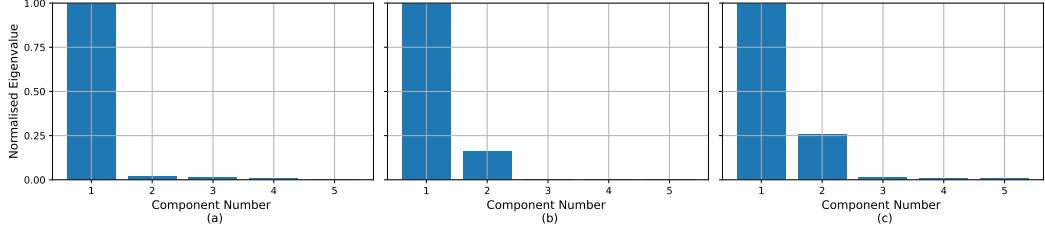

Figure 11: **Ablation analysis of cyclic coordinate loss.** We show the PCA eigenvalue analysis of the learned latent space for a two-body system. The eigenvalues are normalised with respect to the largest component. For a two-body system, **(a)** shows the original SPS-GAN learned representation with $\lambda_{\text{cyclic}} = 0.03$; **(b)** shows SPS-GAN learned representation with $\lambda_{\text{cyclic}} = 0.01$; **(c)** shows SPS-GAN learned representation with $\lambda_{\text{cyclic}} = 0$. SPS-GAN is no longer able to learned the most compact representation with decreasing cyclic coordinate loss weight.

When comparing trajectory with rollout length 30 and $\Delta t = 0.05$, the MSE between the predicted and ground truth trajectory dropped from $3.42 \times 10^{-3}$ at $\lambda_{\text{cyclic}} = 0.03$ to $3.75 \times 10^{-3}$ at $\lambda_{\text{cyclic}} = 0.01$ to $4.40 \times 10^{-2}$ at $\lambda_{\text{cyclic}} = 0$.

## C.2 ABLATION STUDY ON HNN MODULE

We perform ablation study by replacing the HNN (Greydanus et al., 2019) with a Gated Recurrent Unit (GRU) (Cho et al., 2014) for the two-body system in Table 6. SPS-GAN with HNN module outperforms the ablated SPS-GAN with GRU significantly.

Table 6: **Ablation analysis of the HNN module.** Comparison between MSE of predicted trajectory of the two-body system using SPS-GAN with HNN and ablated SPS-GAN with GRU on the latent space. SPS-GAN with HNN module outperforms the ablated SPS-GAN with GRU significantly.

| Timestep | 5 | 10 | 20 | 25 | 30 |
|---|---|---|---|---|---|
| SPS-GAN-HNN | $5.25 \times 10^{-6}$ | $1.02 \times 10^{-4}$ | $1.29 \times 10^{-3}$ | $2.81 \times 10^{-3}$ | $3.68 \times 10^{-3}$ |
| SPS-GAN-GRU | $2.32 \times 10^{-3}$ | $1.33 \times 10^{-2}$ | $6.53 \times 10^{-2}$ | $1.07 \times 10^{-1}$ | $1.60 \times 10^{-1}$ |

As SPS-GAN-traj uses a light weight MLP for decoding the learned latent features into Cartesian coordinates, a non-minimal latent space challenge the expressivity of the decoder, leading to worsened accuracy for the generated trajectory.

## C.3 ABLATION STUDY ON NON-HAMILTONIAN SYSTEM

We study the effect of non-Hamiltonian systems on the performance of SPS-GAN. The damped pendulum system is non a Hamiltonian system as it dissipates energy. The dynamics are governed by

$$\ddot{\theta} = -(d/L)\dot{\theta} - (g/L)\sin\theta, \tag{43}$$

with mass $m = 0.5$, length $L = 1.0$, damping $d = 0.5$, and gravity $g = 3.0$. The damped pendulum can be trivially shown to be a 1-dimensional system. However the minimal learned latent space shows a 3-dimensional space in Figure 12.

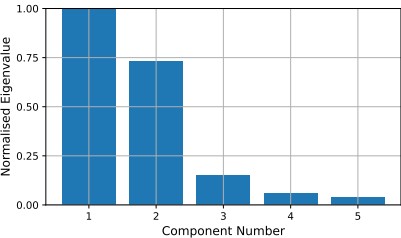

Figure 12: **Ablation analysis of non-Hamiltonian system.** The PCA eigenvalue analysis for the learned latent space of a damped pendulum system is shown. When modelling non-Hamiltonian systems with SPS-GAN, a non-minimal latent space is learned.

## D TRAINING DETAILS

All traning of SPS-GAN was performed on cluster nodes each with 1 Nvidia L40 GPU, 16 core partition of an Intel Xeon Gold 5320, and 64G of DDR4 3200MHz RDIMM.

**SPS-GAN-traj training hyperparmeter.** SPS-GAN for generating time-series data is trained through 50000 epochs with batch size of 128 (pendulum, double pendulum, mass-spring, two-body) or 160 (three-body). It is optimised using Adam (Kingma, 2014) with $\beta_1 = 0.3$ and $\beta_2 = 0.999$ using a learning rate of $5e-5$. The configuration space map and HNN is initialised as an MLP with hidden size 100 using ReLu activation. The trajectory generator is initialised as an MLP with with hidden size 512 using Softplus activation. We set the latent motion space to $d_{\text{lat}} = 20$, content space to $d_{\text{cont}} = 50$, and output size $d_{\text{output}} = 10$.

**SPS-GAN-vid training hyperparmeter.** SPS-GAN for generating video data is trained through 50000 epochs with batch size of 16. It is optimised using Adam with $\beta_1 = 0.3$ and $\beta_2 = 0.999$ using a learning rate of $5e-5$. The configuration space map and HNN is initialised as an MLP with hidden size 100 using ReLu activation. The image generator is initialised as an CNN with 32 filter and 3 channel. We set the latent motion space to $d_{\text{lat}} = 20$, content space to $d_{\text{cont}} = 50$.

