# OpenReview forum: "Physically Plausible Multi-System Trajectory Generation and Symmetry Discovery"
_ICLR.cc/2026/Conference — Submitted to ICLR 2026_

### Official Review · Reviewer_f7by · 2025-10-30

**Soundness:** 3
**Presentation:** 3
**Contribution:** 2
**Rating:** 6
**Confidence:** 3

**Summary:**

The author proposes a novel framework to integrate HNN to sturcture the latent space for physically plausible trajectory generation and video generation.

**Strengths:**

The paper is well-written. The motivation is clear, and the model structure is easy to understand.

**Weaknesses:**

1. The model may not preserve the conservation law in the raw trajectory space.
2. The experiments are insufficient.

**Questions:**

1.	The paper employs Hamiltonian neural network. What if the system doesn’t satisfy the conservation law?
2.	What’s the motivation for generation for multiple systems? Why is studying multiple systems together important?
3.	Some recent papers of learning symmetries in the latent space may be considered, e.g.,
a.	Li, Haoran, et al. "Latent Mixture of Symmetries for Sample-Efficient Dynamic Learning." arXiv preprint arXiv:2510.03578 (2025).
4.	As the latent space dynamics are governed by HNN and the decoder MLP may be nonlinear, how to guarantee that the Hamiltonian structure is preserved in the trajectories?
5.	The experiments are insufficient. There is no ablation study and sensitivity analysis.
6.	The results in Table 1 show that for several systems, the improvement is limited.

---

> ### Author Response · Authors · 2025-11-23
> **Rebuttal Response to Reviewer f7by (1/2)**
>
> > The paper employs Hamiltonian neural networks. What if the system doesn’t satisfy the conservation law?
>
> We test SPS-GAN on modelling non-Hamiltonian systems in Appendix C.3, where the learned latent space no longer matches the analytical understanding of the system being modelled. Specifically, we consider a damped pendulum with $d=0.5$. In this case, SPS-GAN learns an intrinsic dimension of three instead of one (see Figure 12). This indicates that when the system is no longer Hamiltonian, the cyclic-coordinate loss and HNN module are not able to recover the intrinsic dimension of the configuration manifold.
>
> > What’s the motivation for generation for multiple systems? Why is studying multiple systems together important?
>
> Designing a conditional model to generate multiple systems has two benefits:
> * The ability to control the generation process [1, 2]. SPS-GAN is able to extend to unseen configurations of physics (such as mass) and color, as demonstrated in Table 3 and Figure 8.
> * The ability to share parameters across systems. SPS-GAN is able to model five systems using the same configuration space map, HNN recurrent module, and decoder, significantly reducing computation and memory requirements.
>
> [1] Mehdi Mirza, and Simon Osindero. "Conditional Generative Adversarial Nets." arXiv, 2014.\
> [2] Yunjey Choi, et al. "StarGAN: Unified Generative Adversarial Networks for Multi-Domain Image-to-Image Translation." CVPR, 2018.
>
> > Some recent papers of learning symmetries in the latent space may be considered, e.g., a. Li, Haoran, et al. "Latent Mixture of Symmetries for Sample-Efficient Dynamic Learning."
>
> Thank you for pointing out this very relavant paper, we have added additional discussions in our related works section. Specifically, Latent MOS achieves more granularity in the discovered symmetries by embedding specific symmetry groups (such as rotation and translation as well as combinations of symmetry groups) on the latent space. The dynamics of the system is updated using a Mixture-of-Experts composed of these symmetry groups.
>
> SPS-GAN differs from Latent MOS as it focuses on recovering the intrinsic dimension of the configuration space through the identification of cyclic coordinates.
>
> > As the latent space dynamics are governed by HNN and the decoder MLP may be nonlinear, how to guarantee that the Hamiltonian structure is preserved in the trajectories?
>
> The latent space is decoded into trajectories through unconstrained networks, meaning the generated trajectory will not strictly satisfy conservation of energy. However, experiment on energy drift (Figure 9) shows that our generated trajectories have very low total energy variance. In fact, even though SPS-GAN is a generative method, the total energy variance for the SPS-GAN is comparable to HNN. Concretely, for an ideal pendulum system, the total energy mean and variance for the SPS-GAN generated trajectory is 3.109258 ±0.109702 and the total energy mean and variance for the HNN predicted trajectory is 3.119986 ±0.119159. Thank you for raising this concern, we have added additional discussion on energy variance in Section 5.1.

---

> ### Author Response · Authors · 2025-11-23
> **Rebuttal Response to Reviewer f7by (2/2)**
>
> > The experiments are insufficient. There is no ablation study and sensitivity analysis.
>
> We now include ablation studies on the cyclic coordinate loss (see Appendix C.1) and the HNN recurrent module (see Appendix C.2). We show both components contribute to the accuracy and physical consistency of our generated trajectories.
>
> In Appendix C.1, we performed an ablation study on the cyclic-coordinate loss applied to the two-body system. We report the accuracy of generated trajectories for $\lambda_{\mathrm{cyclic}} \in \\{0.03, 0.01, 0\\}$. The last case corresponds to an ablation of the cyclic-coordinate loss. Our experiments show that the accuracy of the generated trajectory deteriorate by an order of magnitude at $\lambda_{\mathrm{cyclic}}=0$ when compared to $\lambda_{\mathrm{cyclic}}=0.03$. The effective dimension of the learned latent space also increases from the correct size of 1 to 2 at both $\lambda_{\mathrm{cyclic}}=0.01$ and $\lambda_{\mathrm{cyclic}}=0$ (see Figure 11). This shows that the cyclic coordinate loss is able to increase generation accuracy while minimizing the latent dimension to match the analytical size.
>
>
> In Appendix C.2, we performed an ablation study on the HNN recurrent module by replacing it with a Gated Recurrent Unit (GRU) [3]. We compare the accuracy of the HNN and GRU versions of SPS-GAN on the two-body system across multiple rollout lengths in Table 5. SPS-GAN with HNN outperformed SPS-GAN with GRU by two orders of magnitude for all rollouts. This shows that the HNN recurrent module is able to produce conditioned trajectories that are easier to decode.
>
> Thank you for pointing out this gap in our original submission. These additional ablation studies verify the contribution of model components and strengthen our manuscript.
>
> [3] Kyunghyun Cho, et al. "Learning Phrase Representations Using RNN Encoder-Decoder for Statistical Machine Translation." EMNLP, 2014.
>
> > The results in Table 1 show that for several systems, the improvement is limited.
>
> We would like to clarify that the core contributions of the proposed model are:
> (1) SPS-GAN can generate physically plausible Cartesian trajectories and videos of dynamical systems (as measured by FVD);
> (2) SPS-GAN can generate videos of dynamical systems with new configurations (e.g., unseen physical parameters and/or particle colors);
> (3) SPS-GAN can identify the dimension of the configuration space without domain knowledge.
>
> We compare SPS-GAN with HNN in Table 1 not to demonstrate improved performance, but comparable performance. This result is not only difficult to achieve with generative models [4,5], but also validates claim (1). Thank you for raising this as a point of confusion. We have clarified the motivation for our approach in our revised manuscript.
>
> [4] Chenyu Zheng, et al. "Revisiting Discriminative vs. Generative Classifiers: Theory and Implications." ICML, 2023.\
> [5] Aleksandar Botev, et al. "Which Priors Matter? Benchmarking Models for Learning Latent Dynamics." NeurIPS, 2021.
>
> **We would like to once again thank you for your time and valuable feedback. Your suggestions has helped us strength our experimental evaluation and related works immensely.**

---

> > ### Comment · Reviewer_f7by · 2025-11-26
> >
> > Thank you for the response. I decided to keep the positive score.

---

### Official Review · Reviewer_WJzx · 2025-10-31

**Soundness:** 2
**Presentation:** 3
**Contribution:** 2
**Rating:** 4
**Confidence:** 3

**Summary:**

The paper introduces Symplectic Phase Space GAN (SPS-GAN), a conditional GAN whose latent dynamics are governed by a Hamiltonian Neural Network (HNN). A configuration-space map projects random motion samples into a latent phase space, where the latent HNN evolves them forward using a leapfrog integrator. Decoders then render either Cartesian trajectories or videos. A cyclic-coordinate loss encourages sparsity in the latent coordinates, interpreted as a form of symmetry constraint. The model is conditioned on system labels and physical parameters to handle multiple systems simultaneously. On five classical benchmarks, it reports trajectory accuracy comparable to supervised HNNs and significantly lower Fréchet distance in video generation.

**Strengths:**

The work integrates HNN dynamics, context-dependent (environment-specific) modeling, and a GAN framework in a coherent way. While each component is known, their synthesis is novel. Also, the paper is clearly written and well organized.

**Weaknesses:**

The claim of “symmetry discovery” is somewhat overstated. The cyclic-coordinate loss captures only a limited form of symmetry, essentially momentum conservation, rather than uncovering more general symmetry groups. Consequently, the most compelling contribution of this paper is its video generation quality under varying system conditions, whereas the symmetry discovery aspect remains nuanced, inferred mainly through latent-space sparsity and t-SNE visualization.

**Questions:**

- Hamiltonian systems often exhibit Hamiltonian bifurcations, which are sudden topological transitions triggered by small parameter variations. A simple example is the Hamiltonian saddle-node bifurcation, $\mathcal{H}(q, p) = p^2/2 - \mu q + q^3/3$, and the Hamiltonian pitchfork bifurcation, $\mathcal{H}(q, p) = p^2 / 2 - \mu (q^2 / 2) + q^4 / 4$, where $\mu \in \mathbb{R}^1$ is a parameter. The later serves as a minimal theoretical framework for single-well ($\mu < 0$) to double-well ($\mu > 0$) transitions often seen in symmetry-breaking theory. It would be interesting to examine whether the proposed model, when trained solely in the pre-bifurcation regime (possibly along with other systems), can predict post-bifurcation behavior.

- The evidence for “correct DOF” is largely t-SNE pictures. t-SNE is not an intrinsic-dimension estimator; its geometry is view-dependent. The physical narratives for 1-DOF two-body and 2-DOF special three-body cases are right, but the validation metric should be more than a plot.

---

> ### Author Response · Authors · 2025-11-23
> **Rebuttal Response to Reviewer WJzx (1/1)**
>
> > Hamiltonian systems often exhibit Hamiltonian bifurcations, which are sudden topological transitions triggered by small parameter variations. It would be interesting to examine whether the proposed model, when trained solely in the pre-bifurcation regime (possibly along with other systems), can predict post-bifurcation behavior.
>
> We now include experiments on the Hamiltonian saddle-node bifurcation [1] in Section 5.3. We report the performance of SPS-GAN trained on pre-bifurcation dynamics and tested on post-bifurcation dynamics. The bifurcation parameter was passed to SPS-GAN as part of the conditional variable $\xi$. The accuracy of the generated pre-bifurcation trajectory (MSE 2.72E-3) was similar to that of post-bifurcation trajectory (MSE 8.83E-3).
>
> Thank you for pointing out this opportunity to assess SPS-GAN’s robustness to variation of system bifurcation.
>
> [1] Wenyang Lyu, et al. "Quantum Dynamics of a One Degree-of-Freedom Hamiltonian Saddle-Node Bifurcation." arXiv, 2021.
>
> > The evidence for “correct DOF” is largely t-SNE pictures. t-SNE is not an intrinsic-dimension estimator; its geometry is view-dependent. The physical narratives for 1-DOF two-body and 2-DOF special three-body cases are right, but the validation metric should be more than a plot.
>
> To support our conclusions quantitatively, we report the magnitude of the first 5 principal components of the learned latent space in Figure 6. The number of dominant principal components using Cattell’s criterion matches with the interpretation of the t-SNE in Figure 5. Furthermore, the cumulative variance paints a similar picture:
> * the first two principal components contribute 76.98% and 19.84% respectively, for a total of 96.82% of the variance for the double pendulum system;
> * the first principal component contributes 94.05% of the variance for the two-body system;
> * the first two principal components contribute 76.64% and 18.61% respectively, for a total of 95.25% of the variance for the three-body system.
>
> Finally, we conducted an experiment on real world pendulum data in Section 5.2, where PCA eigenvalue analysis for the learned latent space shows that SPS-GAN is able to recover a 1-dimensional space.
>
> Thank you for pointing out this concern, and we hope these additional quantitative results are able to alleviate your concern regarding our conclusions in Section 5.1.
>
> **We would like to once again thank you for your time and valuable suggestions. Your suggestions has helped us strength our experimental evaluation immensely.**

---

### Official Review · Reviewer_TFRA · 2025-11-04

**Soundness:** 2
**Presentation:** 2
**Contribution:** 2
**Rating:** 4
**Confidence:** 3

**Summary:**

This paper proposes Symplectic Phase Space GAN (SPS-GAN), a framework designed to capture dynamical system behavior and generalize to unseen physical parameters. The model integrates a Hamiltonian neural network (HNN) recurrent module within a conditional GAN backbone to promote physically plausible generation. Additionally, the authors introduce a modified conditional time-series GAN objective with a physically motivated regularization term that encourages sparse structure in the learned configuration space.

**Strengths:**

* The idea of promoting a sparse representation in the configuration space to simplify the underlying motion dynamics is well-motivated and appears to be effectively implemented.
* The proposed SPS-GAN demonstrates superior empirical performance, outperforming several baseline methods across the evaluated tasks.

**Weaknesses:**

* The paper lacks a clear justification for the motivation of using a generative model (GAN) to represent dynamical trajectories. Given that Newtonian N-body systems evolve deterministically from initial conditions, the necessity of a probabilistic generative approach—as opposed to a deterministic supervised model—is not sufficiently motivated. This concern is compounded by the observation that SPS-GAN does not consistently outperform a supervised HNN baseline, raising questions about the added value of the GAN framework.

* The experimental scope is limited: The primary empirical validation is conducted on two relatively simple, low-dimensional toy benchmarks. It remains unclear whether the proposed method would scale effectively or retain its advantages in more complex, real-world physical systems, which limits assessment of its practical impact.

* The paper does not include a thorough ablation study to disentangle the individual contributions of key components—such as the HNN recurrent module, the GAN backbone, and the sparsity-inducing regularization. Without this, it is difficult to attribute the performance gains to any specific technique of the proposed architecture.

**Questions:**

* The authors use t-SNE visualizations (Figure 5) to justify the dimensionality of the learned latent space. Could they provide a more quantitative or principled explanation for how the size of the latent dimension was determined?
* Could the authors verify the presentation of the symplectic Leapfrog integration in Equations (4)-(6)? These steps do not seem to fully align with the canonical update rules.

**Details Of Ethics Concerns:**

No ethics concerns observed.

---

> ### Author Response · Authors · 2025-11-23
> **Rebuttal Response to Reviewer TFRA (1/2)**
>
> > The paper lacks a clear justification for the motivation of using a generative model (GAN) to represent dynamical trajectories...the necessity of a probabilistic generative approach...is not sufficiently motivated.
>
> While we evaluate the utility of SPS-GAN on physically plausible trajectory prediction (Table 1, Figure 3 and 4), the motivation for using a conditional generative model is that it allows us to learn the shape of the configuration space from the data (Figure 5, Figure 6, and Table 5) and extend to unseen configurations of physics and color (Figure 8 and Table 3). While the former is entirely possible with a discriminative model (e.g., HNN or LNN), the latter two is not.
>
> Learning the configuration manifold allows us to generate new trajectories, and determine the intrinsic dimension of the system without prior domain knowledge. Contrast this to HNN [1] and LNN [2] which requires prior knowledge of the configuration space (in the form of generalized coordinates) to obtain initial conditions to predict trajectories.
>
> Conditioning the generative process on system labels and parameters allows us to extend the generated trajectories to unseen configurations [3, 4]. Compare this to HGN [5] or HGAN [6], where the model is only able to generate new video sequences with similar physical and visual properties.
>
> [1] Samuel Greydanus, et al. "Hamiltonian Neural Networks." NeurIPS, 2019.\
> [2] Miles Cranmer, et al. "Lagrangian Neural Networks." arXiv, 2020.\
> [3] Mehdi Mirza, and Simon Osindero. "Conditional Generative Adversarial Nets." arXiv, 2014.\
> [4] Yunjey Choi, et al. "Stargan: Unified Generative Adversarial Networks for Multi-Domain Image-to-Image Translation." CVPR, 2018.\
> [5] Peter Toth, et al. "Hamiltonian Generative Networks." ICLR, 2020.\
> [6] Christine Allen-Blanchette. "Hamiltonian GAN." L4DC, 2024.
>
> > ...SPS-GAN does not consistently outperform a supervised HNN baseline, raising questions about the added value of the GAN framework.
>
> We would like to clarify that the contributions of SPS-GAN are:
> (1) we can generate physically plausible Cartesian trajectories and videos of dynamical systems (as measured by FVD);
> (2) we can generate videos of dynamical systems with new configurations (e.g., unseen physical parameters and/or particle colors);
> (3) we can identify the dimension of the configuration space without domain knowledge.
>
> We compare with HNN in Table 1 not to demonstrate improved performance, but comparable performance. This result is not only difficult to achieve with generative models [7,8], but also validates claim (1). Thank you for raising this as a point of confusion. We have clarified the motivation for our approach in our revision.
>
> [7] Chenyu Zheng, et al. "Revisiting Discriminative vs. Generative Classifiers: Theory and Implications." ICML, 2023.\
> [8] Aleksandar Botev, et al. "Which Priors Matter? Benchmarking Models for Learning Latent Dynamics." NeurIPS, 2021.
>
> > The primary empirical validation is conducted on two relatively simple, low-dimensional toy benchmarks. It remains unclear whether the proposed method would scale effectively or retain its advantages in more complex, real-world physical systems...
>
> We have expanded our empirical analysis to include a real-world system, the real pendulum [9] (see Section 5.2), and a more complex Hamiltonian system, the Hamiltonian saddle-node bifurcation [10] (see Section 5.3).
>
> In Section 5.2, SPS-GAN trained on real pendulum data is able to generate trajectories with accuracy on par with simulated systems (see Table 1). Applying PCA to the learned latent space shows that SPS-GAN is able to recover the intrinsic dimension of the system (see Figure 7).
>
> In Section 5.3, SPS-GAN was tasked with generating trajectories of post-bifurcation dynamics when trained only on pre-bifurcation dynamics. The accuracy of trajectories generated post-bifurcation (MSE 2.72E-3) was on par with trajectories generated pre-bifurcation (MSE 8.83E-3).
>
> Our manuscript also includes experiments studying:
> * the accuracy of low-dimensional trajectories on five systems (mass-spring, ideal pendulum, chaotic pendulum, two body, and three body systems) in Table 1, Figure 3, and 4;
> * the dimension of t-SNE projections and PCA eigenvalue analysis for multi-body systems (chaotic pendulum, two body, and three body system);
> * the consistency of videos generated on five systems (mass-spring, ideal pendulum, chaotic pendulum, two body, and three body systems), under both constant and variable physical and visual parameters.
>
> We thank you for the opportunity to demonstrate the ability of SPS-GAN to model real world systems and systems exhibiting complex behavior. We have included additional discussion on these new results in our revision.
>
> [9] Michael Schmidt and Hod Lipson. "Distilling Free-Form Natural Laws from Experimental Data." Science, 2009.
> [10] Wenyang Lyu, et al. "Quantum Dynamics of a One Degree-of-Freedom Hamiltonian Saddle-Node Bifurcation." arXiv, 2021.

---

> ### Author Response · Authors · 2025-11-23
> **Rebuttal Response to Reviewer TFRA (2/2)**
>
> > The paper does not include a thorough ablation study to disentangle the individual contributions of key components—such as the HNN recurrent module, the GAN backbone, and the sparsity-inducing regularization. Without this, it is difficult to attribute the performance gains to any specific technique of the proposed architecture.
>
> We now include ablation studies on the cyclic coordinate loss (see Appendix C.1) and the HNN recurrent module (see Appendix C.2). We show both components contribute to the accuracy of our generated trajectories. We do not ablate the GAN backbone as it is what allows us to learn the configuration space, generate novel trajectories, and extend to unseen configurations (please see our response to Question 1).
>
> In Appendix C.1, we performed an ablation study on the cyclic-coordinate loss applied to the two-body system. We report the accuracy of generated trajectories for $\lambda_{\mathrm{cyclic}} \in \\{0.03, 0.01, 0\\}$. The last case corresponds to an ablation of the cyclic-coordinate loss. Our experiments show that the accuracy of the generated trajectory deteriorate by an order of magnitude at $\lambda_{\mathrm{cyclic}}=0$ when compared to $\lambda_{\mathrm{cyclic}}=0.03$. The effective dimension of the learned latent space also increases from the correct size of 1 to 2 at both $\lambda_{\mathrm{cyclic}}=0.01$ and $\lambda_{\mathrm{cyclic}}=0$ (see Figure 11). This shows that the cyclic coordinate loss is able to increase generation accuracy while minimizing the latent dimension to match the analytical size.
>
>
> In Appendix C.2, we performed an ablation study on the HNN recurrent module by replacing it with a Gated Recurrent Unit (GRU) [11]. We compare the accuracy of the HNN and GRU versions of SPS-GAN on the two-body system across multiple rollout lengths in Table 5. SPS-GAN with HNN outperformed SPS-GAN with GRU by two orders of magnitude for all rollouts. This shows that the HNN recurrent module is able to produce conditioned trajectories that are easier to decode.
>
> Thank you for pointing out this omission in our original submission. The addition of these ablation studies clarify the importance of model components and strengthen our paper.
>
> [11] Kyunghyun Cho, et al. "Learning Phrase Representations Using RNN Encoder-Decoder for Statistical Machine Translation." EMNLP, 2014.
>
> > The authors use t-SNE visualizations (Figure 5) to justify the dimensionality of the learned latent space. Could they provide a more quantitative or principled explanation for how the size of the latent dimension was determined?
>
> To provide quantitative support for our conclusions, we have added an PCA eigenvalue analysis of the learned latent space in Figure 6 using Cattell’s criterion. Additionally the results of this analysis indicate that:
> * the first two principal components contribute 76.98% and 19.84% respectively, for a total of 96.82% of the variance for the double pendulum system;
> * the first principal component contributes 94.05% of the variance for the two-body system;
> * the first two principal components contribute 76.64% and 18.61% respectively, for a total of 95.25% of the variance for the three-body system.
>
> These results match both our analytical understanding given in Appendix B.2 and analysis of the t-SNE projection in Figure 5.
>
> Thank you for encouraging additional analysis, these results further support our conclusions.
>
> > Could the authors verify the presentation of the symplectic Leapfrog integration in Equations (4)-(6)? These steps do not seem to fully align with the canonical update rules.
>
> Thank you for pointing this out. We have corrected the update equations for symplectic leapfrog integration.
>
> **We would like to once again thank you for your time and valuable comments. Your review has encouraged us to improve our experimental evaluation and clarify our motivation immensely.**

---

> > ### Comment · Reviewer_TFRA · 2025-11-28
> >
> > Thank the authors for the further responses, which have resolved several of my concerns. Accordingly, I will be raising the rating to 6.

---

### Official Review · Reviewer_c2kG · 2025-11-05

**Soundness:** 3
**Presentation:** 2
**Contribution:** 2
**Rating:** 2
**Confidence:** 4

**Summary:**

This paper proposes SPS-GAN, a GAN-based approach which generates trajectories governed by a Hamiltonian Neural Network. From sampled initial latents, dynamics are generated using HNN with leapfrog integration. The methodology introduces a cyclic-coordinate loss to learn an interpretable system with minimal degrees of freedom. Experiments on synthetic hamiltonian systems demonstrate superiority of SPS-GAN in generating plausible video trajectories.

**Strengths:**

- Generating trajectories with hamiltonian neural networks is an interesting problem.
- The combination of HNN and GANs look promising where SPS-GAN shows superiority in video generation.

**Weaknesses:**

**Novelty appears to be incremental**

I am not sure I can distinguish many differences between this paper and HGAN [1], which is already cited. HGAN is based on HNN and GANs, and introduces the same cyclic-coordinate loss terms as described in this paper (Equation 4 in their paper). While compared to in experiments, I cannot see HGAN discussed in related work.
 - The paper would substantially improve with a discussion highlighting the differences between both methodologies, and further addressing these points with ablation experiments on the improvements.

**Clarity and presentation issues**

- Equations (4-6) have typos. Please revise.
- Section 4 needs to be improved in terms of clarity. In its current stage it is a bit confusing.
  -  For example, it starts by introducing a function f, then it introduces a regularisation term, and it goes back to the function f. I would introduce the regularisation term after defining the final GAN loss.
  - \xi as an argument of the function is explained in Line 202, yet \xi appears first in line 189.
  - If the motion sample has a specific distribution, please indicate it where you define the motion sample.
  - Section 4 might benefit by explaining the overall idea of SPS-GAN together by referencing the elements in Figure 1.
  - Figure 1 and Figure 2 could be grouped together as one.
- Some referenced figures are not aligned in text, making explanations with figures difficult to follow (e.g., when describing results from Figure 3, 4, or 5).

**Experiments section might need revision**
- The MSE results compute grountruth vs predicted trajectory. What is the length of this rollout? What is the MSE in terms of different rollout lengths? An interesting advantage to observe for a Hamiltonian system would be to test if it can generate large rollouts which are plausible.
- For videos, it would be interesting to report results for long/short rollouts and compare.
- Line 305: “Furthermore, the generated trajectory has energy drift within 2.5% of the ground truth conserved energy” what does this imply?
- The predicted trajectories only compare to HNN from 2019. Would it be possible to try other more recent baselines? e.g. HALO or HGN mentioned in related work, HGAN, or a non-hamiltonian continuous-time sequence model (latent ODE).
- Line 307: “We further evaluate SPS-GAN on generating all five systems simultaneously, with explicit system labels and physical parameters used as the conditioning inputs”. Can you provide details of this setup in the main text? Does this mean you provide the model auxiliary labels to distinguish samples from different datasets, and each label activates different \xi’s? It is not clear from this explanation.
- Line 313: “A core contribution of SPS-GAN is the ability to identify symmetries by minimising the dimension of the latent motion space.” The words “latent motion space” have not been used previously and could cause confusion to what they refer to. If they refer to the dimensionality of q, accompany it with the corresponding notation.
- The T-SNE plot to show that the intrinsic dimensionality of the learned systems corresponds to the degrees of freedom is not very intuitive. A quantitative approach to determine intrinsic dimensionality would substantially improve the explanation. In the figure, I cannot see any differences between a and c, yet the authors interpret a has 1 dimension, and c has 2 dimensions.
- How do we select \lambda_{cyclic} in practice? Ablation experiments on how to select \lambda_{cyclic} should be provided.


**Limitations**

The system-specific binary mask requires domain knowledge, which limits applicability to real-world data.


[1] Allen-Blanchette, Christine. "Hamiltonian gan." 6th Annual Learning for Dynamics & Control Conference. PMLR, 2024.

**Questions:**

- If your dynamics are hamiltonian systems, why does the discriminator use an RNN, and not some specific hamiltonian system embedding? This might be an interesting innovation point to explore.
- If we plan to launch this method in real-world systems, how can we design the binary mask?
- What are the benefits from training all the datasets at once if we require additional labels to condition the system parameters? Please correct if I understood this part of the setup wrong.

---

> ### Author Response · Authors · 2025-11-23
> **Rebuttal Response to Reviewer c2kG (1/3)**
>
> > I am not sure I can distinguish many differences between this paper and HGAN [1]...The paper would substantially improve with a discussion highlighting the differences between both methodologies...
>
> While SPS-GAN builds upon some aspects of HGAN, it differs from HGAN architecturally in many ways:
> * SPS-GAN is a conditional model that is able to model different systems with a single network;
> * SPS-GAN introduces a low-dimensional derivative that is able to generate Cartesian trajectories with accuracy on par with supervised HNN;
> * SPS-GAN  is able to extend its generation of videos to unseen configurations of physical properties and color information;
>
> We have added a discussion on HGAN in the related works section of our revision. Thank you for pointing this out.
>
> > Equations (4-6) have typos. Please revise.
>
> Thank you for pointing out this typo. We have updated the equations in the text.
>
> > Section 4 needs to be improved in terms of clarity. In its current stage it is a bit confusing.
>
> We have updated Section 4 in the manuscript. In our revised version, we concretely define the sampling distribution for both the motion and content vector (both are unit Gaussian distributions); we define $\xi$ when it first appears in text in Section 4.1; and we rearrange the subsections so that the cyclic coordinate loss is introduced after the latent space motion model. Changes to our manuscript are highlighted in blue.
>
> > Some referenced figures are not aligned in text, making explanations with figures difficult to follow.
>
> Thank you for your feedback on the readability of our manuscript. We have updated the locations of Figures 3, 4, 5, and 8, as well as Table 1, 2, and 3 to be more closely aligned with their description in text.
>
> > The MSE results compute ground truth vs predicted trajectory. What is the length of this rollout? What is the MSE in terms of different rollout lengths? An interesting advantage to observe for a Hamiltonian system would be to test if it can generate large rollouts which are plausible.
>
> We compute the MSE between ground truth and predicted trajectories on rollouts of 30 timesteps ($\Delta t=0.05$). We now include additional experiments for rollouts of 5, 10, 20, 25, and 30 timesteps (see Table 4 of Appendix B.4) (N.B., the time horizons we report exceed those reported in other generation pipelines with physics based inductive biases [2]). In each experiment, SPS-GAN maintains physical plausibility, performing similarly to the deterministic model HNN [1].
>
> [1] Samuel Greydanus, et al. "Hamiltonian Neural Networks." NeurIPS, 2019.\
> [2] ​​Steindor Saemundsson, et al. "Variational Integrator Networks for Physically Structured Embeddings." AISTATS, 2020.
>
> > For videos, it would be interesting to report results for long/short rollouts and compare.
>
> We agree this would be interesting. We are running these experiments, but they are computationally intensive. We will report results later in the rebuttal period if time allows.
>
> > Line 305: “Furthermore, the generated trajectory has energy drift within 2.5% of the ground truth conserved energy” what does this imply?
>
> The experiment on energy drift (Figure 9) shows that our generated trajectories have very low total energy variance. This implies conservation of energy in the learned Hamiltonian. In fact, even though SPS-GAN is a generative method, the total energy variance for the SPS-GAN is comparable to HNN! Concretely, the total energy mean and variance for the SPS-GAN generated trajectory is 3.109258±0.109702 and the total energy mean and variance for the HNN predicted trajectory is 3.119986±0.119159. Thank you for pointing this out, we have added additional discussion in Section 5.1.

---

> ### Author Response · Authors · 2025-11-23
> **Rebuttal Response to Reviewer c2kG (2/3)**
>
> > The predicted trajectories only compare to HNN from 2019. Would it be possible to try other more recent baselines? e.g. HALO or HGN mentioned in related work, HGAN, or a non-hamiltonian continuous-time sequence model (latent ODE).
>
> HGAN, HALO and HGN are specifically designed for video/image inputs, and therefore expect inputs with structure HxWx3. The dimensionality of the trajectory data used in our setting varies and is therefore incompatible with these networks. NeuralODE [3] and LatentODE [4], however, are easily adapted to our setting and we have included comparisons to these approaches in Table 1. We find that SPS-GAN performs on par with these deterministic models, which are generally expected to have greater accuracy [5, 6, 7]!
>
> [3] Ricky TQ Chen, et al. "Neural Ordinary Differential Equations." NeurIPS, 2018.\
> [4] Yulia Rubanova, et al. "Latent Ordinary Differential Equations for Irregularly-Sampled Time Series." NeurIPS, 2019.\
> [5] Chenyu Zheng, et al. "Revisiting Discriminative vs. Generative Classifiers: Theory and Implications." ICML, 2023.\
> [6] Aleksandar Botev, et al. "Which Priors Matter? Benchmarking Models for Learning Latent Dynamics." NeurIPS, 2021.\
> [7] Partha Ghosh and Mehdi S. M. Sajjadi, et al. "From Variational to Deterministic Autoencoders." ICLR, 2019.
>
> > Line 307: “We further evaluate SPS-GAN on generating all five systems simultaneously, with explicit system labels and physical parameters used as the conditioning inputs”. Can you provide details of this setup in the main text? Does this mean you provide the model auxiliary labels to distinguish samples from different datasets, and each label activates different \xi’s? It is not clear from this explanation.
>
> We use the conditional variable $\xi$ to capture the physical and visual characteristics of the data. Specifically $\xi$ is the concatenation of a categorical variable representing the system label, and 10 random variables representing the physical parameters of the system (e.g., mass). Different system labels will result in a different configuration space map and different dynamics in the HNN recurrent module. We have clarified the role of $\xi$ in our revision.
>
> > Line 313: “A core contribution of SPS-GAN is the ability to identify symmetries by minimising the dimension of the latent motion space.” The words “latent motion space” have not been used previously and could cause confusion to what they refer to. If they refer to the dimensionality of q, accompany it with the corresponding notation.
>
> Thank you for pointing out this typo. This should simply read “latent space”, which represents the motion governed by the HNN module. We have updated this error in our revision.
>
> > The T-SNE plot to show that the intrinsic dimensionality of the learned systems corresponds to the degrees of freedom is not very intuitive. A quantitative approach to determine intrinsic dimensionality would substantially improve the explanation.
>
> We have added an PCA eigenvalue analysis on the latent space produced by SPS-GAN to complement the t-SNE plot in our paper. In our eigenvalue analysis, the double pendulum and three-body systems have 2 dominant eigenvalues while the two-body system has 1. This matches the analytical understanding of these systems as well as the intuition from the t-SNE plot. We have included this analysis using Cattell's criterion in our revision (see Figure 6 and Table 5).
>
> Additionally the cumulative variance shows that:
> * the first two principal components contribute 76.98% and 19.84% respectively, for a total of 96.82% of the variance for the double pendulum system;
> * the first principal component contributes 94.05% of the variance for the two-body system;
> * the first two principal components contribute 76.64% and 18.61% respectively, for a total of 95.25% of the variance for the three-body system.
>
> Thank you for pushing us to conduct these additional analysis, these results further support our conclusions.
>
> > In the figure, I cannot see any differences between a and c, yet the authors interpret a has 1 dimension, and c has 2 dimensions.
>
> Thank you for pointing out this typo - (a) should correspond to the double pendulum and (c) should correspond to the three-body system, both of which are two-dimensional systems. This is corrected in our revision.

---

> ### Author Response · Authors · 2025-11-23
> **Rebuttal Response to Reviewer c2kG (3/3)**
>
> > How do we select \lambda_{cyclic} in practice? Ablation experiments on how to select \lambda_{cyclic} should be provided.
>
> We now include a study on the regularization parameter lambda. In this study, we consider the 2-body system and observe that decreasing lambda or setting it to 0 yields a latent space with 2 dimensions instead of 1. We have included these results in our revision (see Figure 10 of the Appendix).
>
> From our experiments the parameter lambda appears to be fairly robust (we use $\lambda_{\mathrm{cyclic}}=0.03$ for all experiments). A systematic approach for selecting lambda could be to first perform a hyperparameter sweep (on lambda), then a t-sne/PCA analysis on the latent representation. The t-sne/PCA analysis provides the effective dimension of the system based on the choice of lambda, and the best lambda would be the smallest lambda associated with the minimal effective dimension.
>
> > The system-specific binary mask requires domain knowledge, which limits applicability to real-world data.
> >If we plan to launch this method in real-world systems, how can we design the binary mask?
>
> We would like to clarify that our approach does not require additional domain knowledge. The system specific binary mask depends only on the number of particles in the system, and this information is readily obtained from the data itself. Applying the mask allows us to use a single model (one with an arbitrarily sized output dimension) to capture the dynamics of multiple systems with varying numbers of particles. This is in contrast to existing approaches [1, 8, 9] which require knowledge of the generalized coordinate dimension a priori.
>
> [1] Samuel Greydanus, et al. "Hamiltonian Neural Networks." NeurIPS, 2019.\
> [8] Miles Cranmer, et al. "Lagrangian Neural Networks." arXiv, 2020.\
> [9] Yaofeng Zhong, et al. "Dissipative SymODEN: Encoding Hamiltonian Dynamics with Dissipation and Control into Deep Learning." ICLR, 2020.
>
> > What are the benefits from training all the datasets at once if we require additional labels to condition the system parameters?
>
> Designing a conditional model to generate multiple systems has two benefits:
> * The ability to control the generation process [10, 11]. SPS-GAN is able to extend to unseen configurations of physics (such as mass) and color, as demonstrated in Table 3 and Figure 8.
> * The ability to share parameters across systems. SPS-GAN is able to model five systems using the same configuration space map, HNN recurrent module, and decoder, significantly reducing the computation requirement.
>
> [10] Mehdi Mirza, and Simon Osindero. "Conditional Generative Adversarial Nets." arXiv, 2014.\
> [11] Yunjey Choi, et al. "Stargan: Unified Generative Adversarial Networks for Multi-Domain Image-to-Image Translation." CVPR, 2018.
>
> > If your dynamics are hamiltonian systems, why does the discriminator use an RNN, and not some specific hamiltonian system embedding?
>
> The rationale behind our discriminator choice is due to the fact that the generated trajectories from SPS-GAN are position only. This limits the ability to enforce any additional Hamiltonian or symplectic constraints on the discriminator. We designed SPS-GAN-traj this way to mirror SPS-GAN-vid, as there is no simple way to embed velocity information in image/video format without computing discrete gradients.
>
> However you raise a very interesting direction of investigating Hamiltonian discriminators. In fact there are existing machine learning architectures that use Hamiltonian parameterization for both the generator and discriminator [12, 13], interestingly both in quantum physics. However, in these cases they have strong domain knowledge to be able to pick approximate Hamiltonian forms. Thank you for pointing out this interesting research problem, we have added further discussion in the future works section.
>
> [12] Leeseok Kim, et al. "Hamiltonian Quantum Generative Adversarial Networks." Physical Review Research, 2024.\
> [13] James E. Steck and Elizabeth C. Behrman. "Quantum Generative Adversarial Networks: Generating and Detecting Quantum Product States." arXiv, 2024.
>
> **We would like to once again thank you for your time and valuable suggestions. Your review has encouraged us to strengthen our paper immensely.**

---

> ### Author Response · Authors · 2025-11-30
> **Result of Video Generation at Different Rollout Lengths**
>
> > For videos, it would be interesting to report results for long/short rollouts and compare.
>
> We compare the video FVD for SPS-GAN trained and tested with different rollout lengths, where SPS-GAN performs similarly across roll out lengths. The performance of SPS-GAN at 60 is on par or lower than the performance of HGAN/HGN at 30 frames. We report the FVD metric below and will add additional discussion in future revision.
>
> | | SPS-GAN-20 | SPS-GAN-30 | SPS-GAN-60 | HGAN | HGN |
> |:-|:-:|:-:|:-:|:-:|:-:|
> | Mass Spring | 35.86 | 25.63 | 54.26 | 45.68 | 385.08 |
> | Ideal Pendulum| 83.15 | 40.57 | 103.70 | 91.64 | 688.12 |
> | Double Pendulum | 60.12 | 24.12 | 49.74 | 73.21 | 33.94 |
> | Two-Body | 88.68 | 87.12  | 101.73 | 105.85 | 830.91 |
> | Three-Body | 108.19 | 89.08 | 435.35 | 1981.10 | 451.40 |

---

### Author Response · Authors · 2025-11-23

We thank all the reviewers for their time and thoughtful feedback. We are encouraged that they found SPS-GAN to have strong performance on empirical evaluations (c2kG, TFRA) as well as a novel and coherent architecture (TFRA, WJzx, f7by). We also acknowledge the shortcomings raised by the reviewers, including limited quantitative analysis on the latent dimension, lack of ablation study, and insufficient system complexity in empirical evaluation. We hope to address these concerns below.

**Reviewers c2kG, TFRA, and WJzx noted that the size of the latent dimension was determined by visually examining the t-SNE projection.** In order to provide a more quantitative metric to determine the size of the latent space learned by SPS-GAN, we performed PCA eigenvalue analysis for the latent space using Cattell’s criterion. The results are shown in Figure 6 and Table 5. Furthermore, the cumulative variance shows that:
* the first two principal components contribute 76.98% and 19.84% respectively, for a total of 96.82% of the variance for the double pendulum system;
* the first principal component contributes 94.05% of the variance for the two-body system;
* the first two principal components contribute 76.64% and 18.61% respectively, for a total of 95.25% of the variance for the three-body system.

These results align with our analysis in Figure 5 and analytical result in Appendix B.2. Thank you for pointing out this opportunity to strengthen the analysis for the latent space of SPS-GAN.


**Reviewers c2kG, TFRA, and f6by pointed out ablation studies will strengthen the stated contribution of SPS-GAN.** We now include ablation studies on the cyclic coordinate loss (see Appendix C.1) and the HNN recurrent module (see Appendix C.2). We show both components contribute to the accuracy of our generated trajectories. Concretely:
* In Appendix C.1, we performed an ablation study on the cyclic-coordinate loss applied to the two-body system. We report the accuracy of generated trajectories for $\lambda_{\mathrm{cyclic}} \in \\{0.03, 0.01, 0\\}$. The last case corresponds to an ablation of the cyclic-coordinate loss. Our experiments show that the accuracy of the generated trajectory deteriorate by an order of magnitude at $\lambda_{\mathrm{cyclic}}=0$ when compared to $\lambda_{\mathrm{cyclic}}=0.03$. The effective dimension of the learned latent space also increases from the correct size of 1 to 2 at both $\lambda_{\mathrm{cyclic}}=0.01$ and $\lambda_{\mathrm{cyclic}}=0$ (see Figure 11). This shows that the cyclic coordinate loss is able to increase generation accuracy while minimizing the latent dimension to match the analytical size.
* In Appendix C.2, we performed an ablation study on the HNN recurrent module by replacing it with a Gated Recurrent Unit (GRU) [1]. We compare the accuracy of the HNN and GRU versions of SPS-GAN on the two-body system across multiple rollout lengths in Table 5. SPS-GAN with HNN outperformed SPS-GAN with GRU by two orders of magnitude for all rollouts. This shows that the HNN recurrent module is able to produce conditioned trajectories that are easier to decode.

**Reviewers c2kG, TFRA, and WJzx noted that the evaluation of SPS-GAN was performed on simulated systems.** We have expanded our empirical analysis to include a real-world system, the real pendulum [2], and a more complex Hamiltonian system, the Hamiltonian saddle-node bifurcation [3]:
* In Section 5.2, SPS-GAN trained on real pendulum data is able to generate trajectories with accuracy on par with simulated systems (see Table 1). Applying PCA to the learned latent space shows that SPS-GAN is able to recover the intrinsic dimension of the system (see Figure 7).
* In Section 5.3, SPS-GAN was tasked with generating trajectories of post-bifurcation dynamics when trained only on pre-bifurcation dynamics. The accuracy of trajectories generated post-bifurcation (MSE 8.83E-3) was on par with trajectories generated pre-bifurcation (MSE 2.72E-3).

We address the remaining reviewer comments and questions below individually. We highlight changes to our original manuscript with blue text.


[1] Kyunghyun Cho, et al. "Learning Phrase Representations Using RNN Encoder-Decoder for Statistical Machine Translation." EMNLP, 2014.\
[2] Michael Schmidt and Hod Lipson. "Distilling Free-Form Natural Laws from Experimental Data." Science, 2009.\
[3] Wenyang Lyu, et al. "Quantum Dynamics of a One Degree-of-Freedom Hamiltonian Saddle-Node Bifurcation." arXiv, 2021.

---

### Author Response · Authors · 2025-12-03
**Summary of Rebuttal Dicussion**

Dear Area Chair

We would like to thank you for the effort in dealing with the unprecedented situation. We present a summary of the rebuttal discussion and changes to the manuscript.

All the reviews we received were incredibly helpful in materially improving our submission. Specifically, we have added experiments and changes to text focusing on:
* **Quantitative analysis of symmetry discovery (c2kG, TFRA, WJzx)**. Concretely we address the concern of t-SNE being an objective measure of the latent dimension by performing PCA eigenvalue analysis and using Cattel’s criterion to determine the number of dominant principal components (Figure 6, Figure7, Figure11, and Table 5), where the new results supports our original conclusion.
* **Ablation study on model components (c2kG, TFRA, f6by)**. Concretely we address the concern of ambiguity of where the performance gain originates by ablating the sparsity inducing cyclic coordinate loss (Appendix C.1), the HNN recurrent module with a GRU (Appendix C.2), and Hamiltonian assumption of input systems (Appendix C.3), where the results show ablation of components result in uninterpretable latent spaces and deteriorated generation accuracy.
* **Additional experiment and clarification on low-dimensional trajectory generation (c2kG, TFRA, WJzx)**. Concretely we address the concern of the performance of SPS-GAN on trajectory generation by adding comparisons to additional baseline methods (Table 1), clarifying performance difference between generative and discriminative models (Reply to TFRA, f7by), adding results of SPS-GAN across different rollout lengths (Table 4), adding results of SPS-GAN on real world systems (Section 5.2), and adding experiments on systems that exhibits bifurcation (Section 5.3). All additional results show that our generative model achieves accuracy on par with discriminative models while learning an interpretable latent space.
* **Clarification on motivation and relation to existing work (c2kG, TFRA, f7by)**. Concretely we add discussions highlighting how a generative model allows SPS-GAN to learn the dimensions of generalised coordinates (Section 4) without a priori domain knowledge (Reply to c2kG), how a conditional model allows SPS-GAN to learn multiple systems with a smaller model (Reply to TFRA,f7by), differences to existing HGAN (Section 2), and relation to Latent-MOS (Section 2).
* **Additional experiments on energy conservation of output trajectory (c2kG, f7by)**. Concretely we address the concern that energy conservation may be hampered by our decoder by showing energy variance comparable to Hamiltonian Neural Network (Section 5.1) and clarifying our choice of a non-symplectic discriminator (Reply to c2kG).
* **Resolving points of confusion in text (c2kG, TFRA)**. Concretely we added discussion on conditional input $\xi$ (Section 4.1), discussion on sampling distributions (Section 4.1, 4.2), , reorganisation of Section 4 to be in line with model progression, reorganisation of tables and images in Section 5, and fix for typo for symplectic integration (Section 3.2).

We believe that we have thoroughly addressed all weaknesses and questions raised by the four reviewers. Our rebuttal earned support from the two reviewers that were able to engage in discussion before the policy change.

We hope this summary is useful for your evaluation and we once again thank you for your valuable time and effort in ensuring a fair and transparent review process.

---

### Meta-Review · Area_Chair_6vY9 · 2026-01-13

**Summary:**

The reviewers acknowledged that the paper proposes a coherent integration of Hamiltonian Neural Networks (HNNs) with a conditional GAN framework and demonstrates strong empirical results, particularly in video generation. However, several substantial concerns consistently emerged across reviews and ultimately informed my recommendation to reject this submission.

First, the novelty and differentiation from closely related prior work remain insufficiently convincing, especially with respect to Hamiltonian GAN (HGAN). Reviewer c2kG explicitly stated that “Novelty appears to be incremental” and expressed difficulty distinguishing the proposed method from HGAN, noting that both approaches rely on HNNs and introduce similar cyclic-coordinate losses. While the authors later clarified architectural differences, the reviewers’ concern that the contribution may amount to a conditional or extended variant of existing methods was not fully dispelled.

Second, the motivation for adopting a GAN-based generative framework for deterministic Hamiltonian systems was initially unclear and remains debatable. Reviewer TFRA questioned the necessity of a probabilistic generative model given that the system dynamics are deterministic, emphasizing that SPS-GAN does not consistently outperform supervised HNN baselines. Although the rebuttal improved the conceptual justification, this concern reflects a broader uncertainty about whether the generative formulation provides fundamentally new capabilities rather than an alternative modeling choice.

Third, claims related to “symmetry discovery” were viewed as overstated. Reviewer WJzx emphasized that the proposed cyclic-coordinate loss primarily captures limited forms of symmetry (e.g., momentum conservation) rather than discovering general symmetry groups, and that much of the evidence relied on latent-space sparsity and visualization. While additional quantitative analyses were introduced, concerns remain about whether the term “symmetry discovery” is fully justified by the methodology and experiments.

Finally, the empirical evaluation, although strengthened during rebuttal, still raises concerns regarding scope and interpretability. Multiple reviewers initially pointed out limitations such as reliance on t-SNE for intrinsic dimensionality estimation, limited ablations, and unclear rollout settings. Although the authors addressed several of these issues, the remaining gaps—particularly regarding the fundamental novelty and strength of the claimed contributions—continue to weigh against acceptance.

**Reviewer Concerns:**

Concerns largely addressed by the rebuttal

- Lack of quantitative support for intrinsic dimensionality / DOF claims:
Reviewers c2kG, TFRA, and WJzx criticized the reliance on t-SNE, noting that “t-SNE is not an intrinsic-dimension estimator” (WJzx).
→ The authors addressed this by adding PCA eigenvalue analyses with Cattell’s criterion and reporting cumulative variance across systems. This constitutes a substantive and appropriate response.

- Insufficient ablation studies:
Reviewers TFRA and f7by highlighted the absence of ablations disentangling the contributions of the cyclic-coordinate loss and the HNN recurrent module.
→ The rebuttal introduced targeted ablations (cyclic loss removal and HNN vs. GRU replacement), showing significant degradation in both accuracy and latent interpretability. These additions adequately addressed this concern.

- Limited experimental scope and rollout clarification:
Reviewer c2kG asked for clarification on rollout lengths and long-horizon behavior.
→ The authors added results for multiple rollout lengths and further experiments on real pendulum data and Hamiltonian saddle-node bifurcation, partially mitigating concerns about overly simplistic benchmarks.

- Technical clarity and correctness:
Several reviewers (c2kG, TFRA) pointed out typos and inconsistencies in the leapfrog integration equations and notation.
→ The authors acknowledged and corrected these issues, improving clarity and correctness.

Concerns that remain outstanding

- Incremental novelty and insufficient differentiation from HGAN:
Reviewer c2kG’s core concern—“I am not sure I can distinguish many differences between this paper and HGAN”—is only partially addressed. While the authors list architectural differences (conditional modeling, multi-system training, measurement-agnostic inputs), the rebuttal does not provide sufficiently strong empirical or conceptual evidence that these differences constitute a fundamentally new contribution rather than an extension of existing frameworks.

- Strength and scope of the “symmetry discovery” claim:
Reviewer WJzx argued that the proposed method captures only limited forms of symmetry and that the claim of symmetry discovery is overstated. Although the authors improved quantitative analysis of latent dimensionality, the broader claim of discovering symmetries—beyond identifying cyclic coordinates—remains only weakly supported.

- Necessity of the GAN formulation:
While Reviewer TFRA was ultimately persuaded by the authors’ explanation that a generative model enables configuration-manifold discovery and generalization to unseen parameters, the concern remains relevant at a broader level. The rebuttal explains why a GAN was chosen, but does not fully establish that GAN-based modeling is essential or uniquely advantageous compared to alternative generative or structured latent approaches.

**Reviewer Scores:**

- Reviewer c2kG (original score: 2 – reject)
This reviewer’s most critical concerns focused on novelty, differentiation from HGAN, and overstated claims. While the rebuttal addressed several technical and experimental issues (e.g., ablations, rollout details, PCA analysis), the central concern regarding incremental novelty was not fully resolved.
  - Expected score change: Likely remains 2,  still below the acceptance threshold.

- Reviewer TFRA (original score: 4 – marginally below acceptance threshold)
Reviewer TFRA explicitly stated in the discussion that the additional responses “resolved several of my concerns” and that they would raise their rating to 6.
  - Expected score change: Increased to 6, as explicitly indicated by the reviewer.

- Reviewer WJzx (original score: 4 – marginally below acceptance threshold)
The rebuttal directly addressed this reviewer’s requests for bifurcation experiments and quantitative DOF analysis, which likely alleviated some concerns. However, skepticism regarding the strength of the symmetry discovery claim likely remains.
  - Expected score change: Modest increase, possibly from 4 to 6, but still around the borderline.

- Reviewer f7by (original score: 6 – marginally above acceptance threshold)
This reviewer’s concerns about conservation laws, non-Hamiltonian systems, and lack of ablations were largely addressed in the rebuttal. However, they already assigned a borderline-positive score and indicated they “would not mind if paper is rejected.”
  - Expected score change: Likely remains around 6, with limited upward movement.

---

### Decision · Program_Chairs · 2026-01-26

Reject